# A genome-wide resource for the analysis of protein localisation in *Drosophila*

Mihail Sarov[1]*, Christiane Barz[2], Helena Jambor[1], Marco Y Hein[3], Christopher Schmied[1], Dana Suchold[1], Bettina Stender[2], Stephan Janosch[1], Vinay Vikas KJ[4], RT Krishnan[4], Aishwarya Krishnamoorthy[4], Irene RS Ferreira[2], Radoslaw K Ejsmont[1], Katja Finkl[2], Susanne Hasse[1], Philipp Kämpfer[5], Nicole Plewka[2], Elisabeth Vinis[1], Siegfried Schloissnig[5], Elisabeth Knust[1], Volker Hartenstein[6], Matthias Mann[3], Mani Ramaswami[7], K VijayRaghavan[4], Pavel Tomancak[1]*, Frank Schnorrer[2]*

[1]Max Planck Institute of Cell Biology and Genetics, Dresden, Germany; [2]Muscle Dynamics Group, Max Planck Institute of Biochemistry, Martinsried, Germany; [3]Department of Proteomics and Signal Transduction, Max Planck Institute of Biochemistry, Martinsried, Germany; [4]Centre for Cellular and Molecular Platforms, National Centre for Biological Sciences, Tata Institute of Fundamental Research, Bangalore, India; [5]Heidelberg Institute of Theoretical Studies, Heidelberg, Germany; [6]Department of Molecular Cell and Developmental Biology, University of California, Los Angeles, Los Angeles, United States; [7]Institute of Neuroscience, Trinity College Dublin, Dublin, Ireland

*For correspondence: sarov@mpi-cbg.de (MS); tomancak@mpi-cbg.de (PT); schnorrer@biochem.mpg.de (FS)

**Abstract** The *Drosophila* genome contains >13000 protein-coding genes, the majority of which remain poorly investigated. Important reasons include the lack of antibodies or reporter constructs to visualise these proteins. Here, we present a genome-wide fosmid library of 10000 GFP-tagged clones, comprising tagged genes and most of their regulatory information. For 880 tagged proteins, we created transgenic lines, and for a total of 207 lines, we assessed protein expression and localisation in ovaries, embryos, pupae or adults by stainings and live imaging approaches. Importantly, we visualised many proteins at endogenous expression levels and found a large fraction of them localising to subcellular compartments. By applying genetic complementation tests, we estimate that about two-thirds of the tagged proteins are functional. Moreover, these tagged proteins enable interaction proteomics from developing pupae and adult flies. Taken together, this resource will boost systematic analysis of protein expression and localisation in various cellular and developmental contexts.

## Introduction

With the complete sequencing of the *Drosophila* genome (*Adams et al., 2000*) genome-wide approaches have been increasingly complementing the traditional single gene, single mutant studies. This is exemplified by the generation of a genome-wide transgenic RNAi library (*Dietzl et al., 2007*) to systematically assess gene function in the fly or by the documentation of the entire developmental transcriptome during all stages of the fly's life cycle by mRNA sequencing (*Graveley et al., 2011*). Furthermore, expression patterns were collected for many genes during *Drosophila* embryogenesis by systematic mRNA in situ hybridisation studies in different tissues (*Hammonds et al., 2013*; *Tomancak et al., 2002*; *2007*). Particularly for transcription factors (TFs), these studies revealed complex and dynamic mRNA expression patterns in multiple primordia and

**eLife digest** The fruit fly *Drosophila melanogaster* is a popular model organism in biological research. Studies using *Drosophila* have led to important insights into human biology, because related proteins often fulfil similar roles in flies and humans. Thus, studying the role of a protein in *Drosophila* can teach us about what it might do in a human.

To fulfil their biological roles, proteins often occupy particular locations inside cells, such as the cell's nucleus or surface membrane. Many proteins are also only found in specific types of cell, such as neurons or muscle cells. A protein's location thus provides clues about what it does, however cells contain many thousands of proteins and identifying the location of each one is a herculean task.

Sarov et al. took on this challenge and developed a new resource to study the localisation of all *Drosophila* proteins during this animal's development. First, genetic engineering was used to tag thousands of *Drosophila* proteins with a green fluorescent protein, so that they could be tracked under a microscope.

Sarov et al. tagged about 10000 *Drosophila* proteins in bacteria, and then introduced almost 900 of them into flies to create genetically modified flies. Each fly line contains an extra copy of the tagged gene that codes for one tagged protein. About two-thirds of these tagged proteins appeared to work normally after they were introduced into flies. Sarov et al. then looked at over 200 of these fly lines in more detail and observed that many of the proteins were found in particular cell types and localized to specific parts of the cells. Video imaging of the tagged proteins in living fruit fly embryos and pupae revealed the proteins' movements, while other techniques showed which proteins bind to the tagged proteins, and may therefore work together in protein complexes.

This resource is openly available to the community, and so researchers can use it to study their favourite protein and gain new insights into how proteins work and are regulated during *Drosophila* development. Following on from this work, the next challenge will be to create more flies carrying tagged proteins, and to swap the green fluorescent tag with other experimentally useful tags.

organs during development (*Hammonds et al., 2013*), supposedly driven by specific, modular enhancer elements (*Kvon et al., 2014*). Furthermore, many mRNAs are not only dynamically expressed but also subcellularly localised during *Drosophila* oogenesis (*Jambor et al., 2015*) and early embryogenesis (*Lécuyer et al., 2007*). Together, these large-scale studies at the RNA level suggest that the activity of many genes is highly regulated in different tissues during development. Since the gene function is mediated by the encoded protein(s), the majority of proteins should display particular expression and subcellular localisation patterns that correlate with their function.

However, a lack of specific antibodies or live visualisation probes thus far hampered the systematic survey of protein expression and localisation patterns in various developmental and physiological contexts in *Drosophila*. Specific antibodies are only available for about 450 *Drosophila* proteins (*Nagarkar-Jaiswal et al., 2015*), and the versatile epitope-tagged UAS-based overexpression collection that recently became available (*Bischof et al., 2013*; *Schertel et al., 2015*) is not suited to study protein distribution at endogenous expression levels. Collections of knock-in constructs are either limited to specific types of proteins (*Dunst et al., 2015*) or rely on inherently random genetic approaches, such as the large-scale protein-trap screens or the recently developed MiMIC (Minos Mediated Insertion Cassette) technology (*Venken et al., 2011*). The classical protein-trap screens are biased for highly expressed genes, and altogether recovered protein traps in 514 genes (*Buszczak et al., 2007*; *Lowe et al., 2014*; *Morin et al., 2001*; *Quiñones-Coello et al., 2007*). The very large-scale MiMIC screen isolated insertions in the coding region of 1862 genes, 200 of which have been converted into GFP-traps available to the community (*Nagarkar-Jaiswal et al., 2015*). Both approaches rely on transposons to mediate cassette insertion and require integration into an intron surrounded by coding exons for successful protein tagging. Thus, about 3,000 proteins, whose ORF is encoded within a single exon, cannot be tagged by these approaches (analysis performed using custom Perl script available here: https://github.com/tomancak/intronless_CDS_analysis). Together, this creates a significant bias towards trapping a particular subset of the more than 13,000 protein coding genes in the fly genome.

Hence, the *Drosophila* community would profit from a resource that enables truly systematic protein visualisation at all developmental stages for all protein-coding genes, while preserving the endogenous expression pattern of the tagged protein. One strategy to generate a comprehensive resource of tagged proteins is to tag large genomic clones by recombineering approaches in bacteria and transfer the resulting tagged clones into animals as third copy reporter allele as was previously done in *C. elegans* (*Sarov et al., 2012*). The third copy reporter allele approach was used successfully in *Drosophila* with large genomic BAC or fosmid clones derived from the fly genome. In *Drosophila,* it is possible to insert this tagged copy of the gene as a transgene at a defined position into the fly genome (*Venken et al., 2006*). It has been shown that such a transgene recapitulates the endogenous expression pattern of the gene in flies and thus likely provides a tagged functional copy of the gene (*Ejsmont et al., 2009*; *Venken et al., 2009*).

Here, we introduce a comprehensive genome-wide library of almost 10000 C-terminally tagged proteins within genomic fosmid constructs. For 880 constructs, covering 826 different genes we generated transgenic lines, 765 of which had not been tagged by previous genetic trapping projects. Rescue experiments using a subset of lines suggest that about two thirds of the tagged proteins are functional. We characterised the localisation patterns for more than 200 tagged proteins at various developmental stages from ovaries to adults by immunohistochemistry and by live imaging. This identified valuable markers for various tissues and subcellular compartments, many of which are detectable in vivo by live imaging. Together, this shows the wide range of possible applications and the potential impact this publically available resource will have on *Drosophila* research and beyond.

## Results

Our goal was to generate a comprehensive resource that allows the investigation of protein localisation and physical interactions for any fly protein of interest through a robust, generic tagging pipeline in bacteria, which is followed by a large-scale transgenesis approach (*Figure 1A*). We based our strategy on a *Drosophila melanogaster* FlyFos library of genomic fosmid clones, with an average size of 36 kb, which covers most *Drosophila* genes (*Ejsmont et al., 2009*). Our two-step tagging strategy first inserts a generic 'pre-tag' at the C-terminus of the protein, which is then replaced by any tag of choice at the second tagging step, for example with a superfolder-GFP (sGFP) tag to generate the sGFP TransgeneOme clone library. These tagged clones are injected into fly embryos to generate transgenic fly-TransgeneOme (fTRG) lines, which can be used for multiple in vivo applications. (*Figure 1A*).

### sGFP TransgeneOme – a genome-wide tagged FlyFos clone library

We aimed to tag all protein coding genes at the C-terminus of the protein, because a large number of regulatory elements reside within or overlap with the start of genes, including alternative promoters, enhancer elements, nucleosome positioning sequences, etc. These are more likely to be affected by a tag insertion directly after the start codon. Signal sequences would also be compromised by an inserted tag after the start codon. This is in agreement with a recently published dataset favouring C-terminal compared to N-terminal tagging (*Stadler et al., 2013*). Additionally, the C-termini in the gene models are generally better supported by experimental data than the N-termini due to an historical bias for 3'-end sequencing of ESTs. Thus, C-terminal tagging is more likely to result in a functional tagged protein than N-terminal tagging, although we are aware of the fact that some proteins will be likely inactivated by addition of a tag to the C-terminus. Moreover, only about 1400 protein coding genes contain alternative C-termini, resulting in all protein isoforms labelled by C-terminal tagging for almost 90% of all protein-coding genes (analysis performed using custom Perl script available here: https://github.com/tomancak/alternative_CDS_ends).

In a series of pilot experiments, we tested the functionality of several tagging cassettes with specific properties on a number of proteins (*Figure 1—figure supplement 1*, *Table 1*). For the genome-wide resource, we applied a two-step tagging strategy, whereby we first inserted a non-functional 'pre-tagging' cassette consisting of a simple bacterial selection marker, which is flanked with linker sequences present in all of our tagging cassettes. This strategy enables a very efficient replacement of the 'pre-tag' by any tag of interest using homologous recombination mediated cassette exchange in bacteria (*Hofemeister et al., 2011*). As fluorescent proteins and affinity tags with improved properties are continuously being developed, specific clones or the entire resource can be

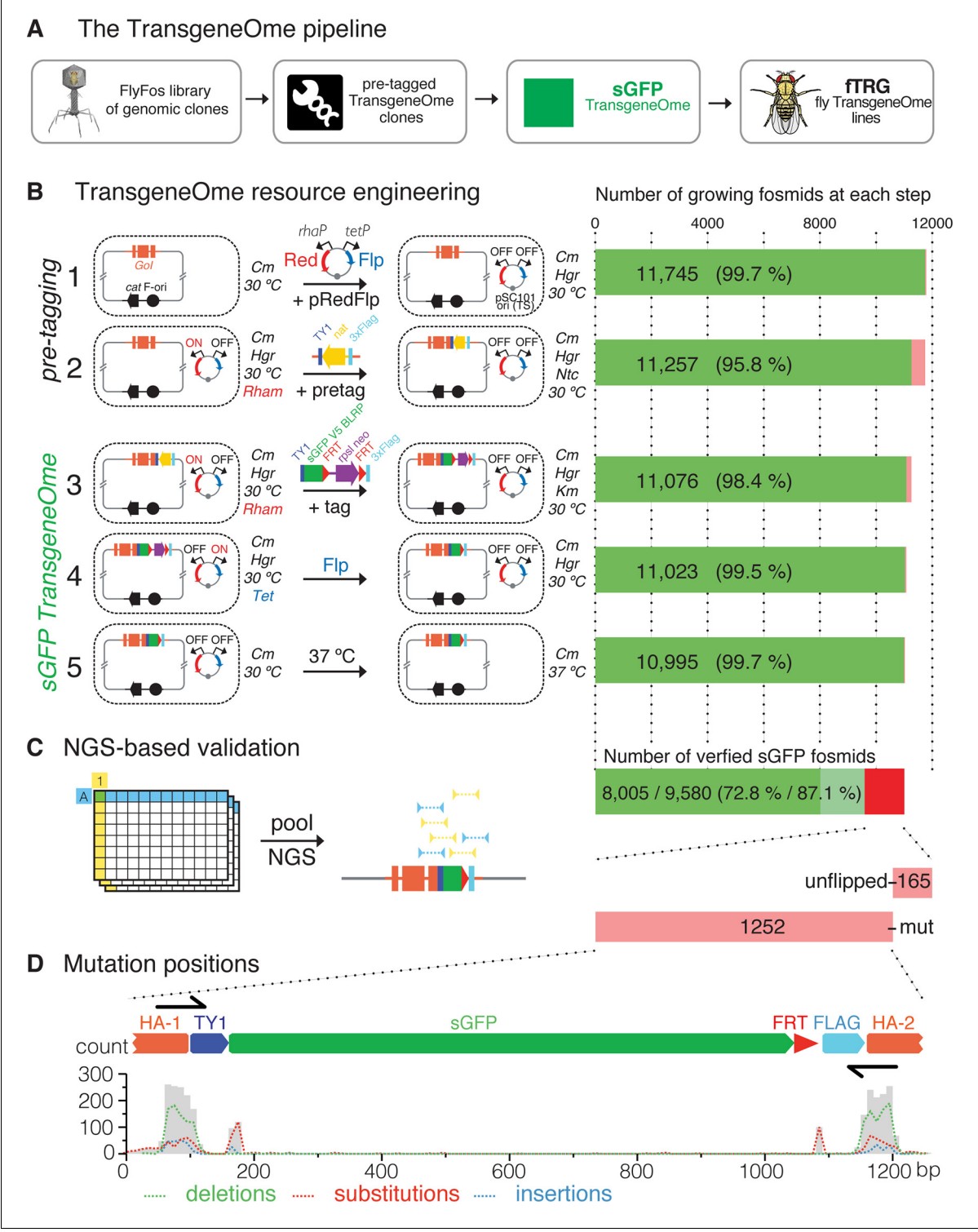

**Figure 1.** Generation of the TransgeneOme library. (**A**) Overview of the tagging strategy. Liquid culture recombineering is used to insert a 'pre-tagging' cassette into FlyFos genomic clones in bacteria. This cassette can then be replaced by a simple, universal, recombineering reaction with any tag of choice, here, a superfolder GFP tag (sGFP) to generate the sGFP TransgeneOme clone library. These clones are transformed into flies generating transgenic FlyFos libraries that can be used for multiple in vivo applications. (**B**) TransgeneOme resource engineering. The steps of the recombineering pipeline are shown on the left with the success rate of each step indicated on the right (red colour denotes bacterial clones that did not grow). The *E. coli* cells are schematically represented with a dotted circle. With the first two steps the 'pre-tagging' cassette is inserted, which is replaced in the next three steps with the sGFP cassette to generate the sGFP TransgeneOme library. See text and methods section for details; abbreviations: *Gol* – gene of

*Figure 1 continued on next page*

*Figure 1 continued*

interest; *cat* – chloramphenicol resistance gene, F-ori – the fosmid vector replication control sequences; pRedFlp – recombineering helper plasmid with the pSC101 temperature sensitive origin of replication, which can be maintained at 30°C and is removed at the final step by temperature shift to 37°C; it carries the Red gbaA operon (Red), which drives homologous recombination in vivo under the control of the L-Rhamnose (Rham) inducible promoter (rhaP) and the Flp recombinase (Flp) under the control of tetracycline (Tet) inducible promoter (tetP); pretag – a pre-tagging cassette consisting of the Nourseothricin resistance gene (nat), flanked by the 2xTY and 3xFlag tag, which provide regions of homology to the tagging cassette (tag), consisting of the TY1 tag, superfolder GFP (sGFP), the V5 tag and the target peptide for the birA biotin ligase (blrp), the FRT-flanked selection/counter-selection operon rpsl-neo (confers streptomycin sensitivity and kanamycin resistance) and the 3xFlag tag. (C) Next-generation-sequencing (NGS)-based validation of the sGFP TransgeneOme library. Schematic of the bar coding (BC) strategy of row and column pools is shown to the left and sequencing results to the right. In this example, a clone with the coordinates A1 will receive the row A barcode (blue) and the column 1 barcode (yellow), which allow the mapping of the NGS sequence reads to the respective well. By using the mate-paired strategy the reads mapping to the tag can be assigned to a specific transgene, i.e. only reads where one mate of the pair maps to the tag and the other to the genome are used. Sequenced tags within fosmids without point mutations are shown in solid green, clones without mutation in tagging cassette but incomplete coverage in light green and clones with mutation(s) or un-flipped cassette are shown in red. (D) Statistics of the mutation distributions with deletions indicated by green, substitutions by red and insertions by blue interrupted lines. Note that most mutations reside within the recombineering primer sequences (denoted as black arrows).

The following figure supplement is available for figure 1:

**Figure supplement 1.** Tagging cassettes.

easily re-fitted to any new tagging cassette. For the genome-scale resource, we selected a tagging cassette suitable for protein localisation and complex purification studies, consisting of the 2xTY1 tag as a flexible linker, the superfolder GFP coding sequence (*Pédelacq et al., 2005*), the V5 tag, followed by specific protease cleavage sites (for the PreScission- and TEV-proteases), the biotin ligase recognition peptide (BLRP) tag allowing for specific in vivo or in vitro biotinylation (*Deal and Henikoff, 2010*; *Vernes, 2014*), and the 3xFLAG tag (*Figure 1—figure supplement 1*).

Of the 13937 protein-coding genes in the dmel5.43 genome assembly, 11787 genes (84.6% ) were covered by a suitable fosmid from the original FlyFos library (*Ejsmont et al., 2009*), extending at least 2.5 kb upstream and 2.5 kb downstream of the annotated gene model. For picking clones, designing oligonucleotides for recombineering, and for tracking all steps of the transgene engineering process, as well as for providing access to all construct sequences and validation data we used the previously developed TransgeneOme database (*Sarov et al., 2012*), which is available online (https://transgeneome.mpi-cbg.de).

For high-throughput tagging of the *Drosophila* FlyFos clones, we developed an improved version of our previously applied high-throughput, 96-well format liquid culture recombineering pipeline (*Ejsmont et al., 2011*; *Sarov et al., 2012*), and we applied it to create a single tagged construct for

**Table 1.** TransgeneOme constructs and fTRG lines - overview of TransgeneOme constructs generated and verified by sequencing for the different pilot sets and the genome-wide set, including the respective numbers of the transgenic fTRG lines generated.

**Tagged constructs and transgenic lines**

|  | constructs | verified constructs | transgenic lines |
| --- | --- | --- | --- |
| 'pre-tagging' - TransgeneOme | 11257 | – | – |
| TY1-sGFP-V5-BLRP-FLAG |  |  | 799 |
| - TransgeneOme | 10995 | 9580 |  |
| - pilot set | 1328 | 1328 |  |
| TY1-T2A-sGFPnls-FLAG |  |  |  |
| - pilot set | 273 | 273 | 30 |
| TY1-sGFP-FLAG |  |  |  |
| - pilot set | 644 | 483 | 51 |
| unique constructs | 23169 | 10711 | 880 |
| unique genes | 11257 | 9993 | 826 |

each gene covered by a suitable fosmid. The high efficiency of recombineering in *E. coli* allowed for multi-step DNA engineering in 96-well format liquid cultures with single clone selection only at the last step. The pipeline consists of five steps (*Figure 1B*). First, the pRedFlp helper plasmid containing all genes required for homologous recombination and the Flippase-recombinase (under L-rhamnose and tetracycline control, respectively) was introduced into *E. coli* by electroporation. Second, the 'pre-tagging' cassette containing a bacterial antibiotic resistance gene was inserted by homologous recombination with gene-specific homology arms of 50 base pairs. Third, the sGFP-V5-BLRP tagging cassette, including an FRT-flanked selection and counter-selection cassette, was inserted to replace the 'pre-tagging' cassette. Since the linker sequences in the 'pre-tagging' cassette are identical to the tagging cassette, the tagging cassette was simply excised from a plasmid by restriction digest and no PCR amplification was required. This strongly reduced the risk of PCR-induced mutations in the tagging cassette. Fourth, the selection marker was excised by the induction of Flippase expression. Fifth, the helper plasmid was removed by suppression of its temperature sensitive replication at 37°C (*Meacock and Cohen, 1980*) and single clones were isolated from each well by plating on selective solid agar plates.

All five steps of the engineering pipeline were highly efficient (between 95.8 and 99.7% ), resulting in an overall efficiency of 93.6% or 10995 growing cultures (*Figure 1B*). To validate the sequence of the engineered clones, we developed a new next-generation-sequencing (NGS)-based approach (*Figure 1C*). In short, we pooled single clones from all 96-well plates into 8 rows and 12 columns pools, prepared barcoded mate pair libraries from each pool, and sequenced them using HiSeq2500 (Illumina). The mate pair strategy allowed us to map the otherwise common tag coding sequence to a specific clone in the library and thus to verify the integrity of the tagging cassette insertion in the clones with single nucleotide resolution (see Materials and methods for details). When applied to the final sGFP TransgeneOme collection, we detected no mutations for 9580 constructs (87.1% ). 8005 (72.8% ) of these clones had complete sequence coverage of the tag-coding sequence and thus represent the most reliable subset of the tagged library (*Figure 1C*). For 1417 of the clones (12.8% ), one or more differences to the expected sequences were detected. The most common differences were point mutations, which cluster almost exclusively to the homology regions in the oligonucleotides used to insert the 'pre-tagging' cassette. This is suggestive of errors in the oligonucleotide synthesis. Another subset of point mutations clustered around the junctions between the homology arms and the rest of the tagging cassette, indicating an imprecise resolving of the homology exchange reaction in small subset of clones (*Figure 1D*). Finally, a small group of clones (165) still contained an un-flipped selection cassette. The NGS results were confirmed by Sanger sequencing of the entire tag-coding sequence for a subset of constructs (*Supplementary file 1*). The detailed sequencing results for all clones are available at https://transgeneome.mpi-cbg.de.

Taken together, the sGFP TransgeneOme and our pilot tagging experiments resulted in 10711 validated tagged clones, representing 9993 different *Drosophila* genes. (*Table 1*, *Supplementary file 1*). The clones are available from Source BioScience as *Drosophila* TransgeneOme Resource (MPI-CBG) (http://www.lifesciences.sourcebioscience.com/clone-products/non-mammalian/drosophila/drosophila-transgeneome-resource-mpi-cbg/). Moreover, the 'pre-tagged' TransgeneOme library is a versatile resource for generating fosmid clones with arbitrary tags at the C-terminus of the gene models.

## Fly TransgeneOme (fTRG) – a collection of fly strains with tagged fosmids transgenes

We next established a pipeline to systematically transform the tagged TransgeneOme clones into flies. To efficiently generate fly transgenic lines, we injected the tagged fosmid constructs into a recipient stock carrying the attP landing site VK00033 located at 65B on the third chromosome using a transgenic *nanos*-ΦC31 source (*Venken et al., 2006*). For some genes positioned on the third chromosome, we injected into VK00002 located on the second chromosome at 28E to simplify genetic rescue experiments. In total, we have thus far generated lines for 880 tagged constructs representing 826 different genes (*Table 1*, *Supplementary file 2*). These genes were partially chosen based on results of a public survey amongst the *Drosophila* community to identify genes for which there is the strongest demand for a tagged genomic transgenic line. 765 (87% ) of the newly tagged genes have not been covered by the previous protein-trap projects (*Supplementary file 2*), hence, these should be particularly useful for the fly community. From our pilot tagging experiments, we

made 51 lines for the 2xTY1-sGFP-3xFLAG tag and 30 lines for the 2xTY1-T2A-sGFPnls-FLAG transcriptional reporter. The majority of the lines (799) were generated with the versatile 2xTY1-sGFP-V5-Pre-TEV-BLRP-3xFLAG tag, used for the genome-wide resource (*Figure 1—figure supplement 1*, *Table 1*). The collection of fly lines is called 'tagged fly TransgeneOme' (fTRG) and all 880 fTRG lines have been deposited at the VDRC stock centre for ordering (http://stockcenter.vdrc.at).

To assess whether the tagged fosmids in our transgenic library are functional, we have chosen a set of 46 well-characterised genes, mutants of which result in strong developmental phenotypes. For most cases, we tested null or strong hypomorphic alleles for rescue of the respective phenotypes (embryonic lethality, female sterility, flightlessness, etc.) with the tagged fosmid lines. More than two-thirds of the lines (31 of 46), including tagged lines of *babo*, *dlg1*, *dl*, *fat*, *Ilk*, *LanB1*, *numb*, *osk*, *rhea*, *sax*, *smo* and *yki* rescued the mutant phenotypes (*Figure 2A*, *Table 2*), demonstrating that the majority of the tagged genes is functional. Our rescue test set is biased towards important developmental regulators; 10 of the 15 genes that did not show a rescue are transcription factors with multiple essential roles during development, such as *esg*, *eya*, *odd*, *sna* and *salm*. Thus, their expression is likely regulated by complex cis-regulatory regions that may not be entirely covered by the available fosmid clone; for example wing-disc enhancers are located more than 80 kb away from the transcriptional start of the *salm* gene (*De Celis et al., 1999*). Hence, we expect that a typical gene, which is embedded within many other genes in the middle of the fosmid clone, is more likely to be functional. Together, these data suggest that both the genome-wide tagged construct library and the transgenic fTRG library provide functional reagents that are able to substitute endogenous protein function.

## Expression of fTRG lines in the ovary

To demonstrate the broad application spectrum of our fly TransgeneOme library, we analysed tagged protein expression and subcellular localisation in multiple tissues at various developmental stages. Germline expression in flies differs substantially from somatic expression, requiring particular basal promoters and often specialised 3'UTRs (*Ni et al., 2011*; *Rørth, 1998*). Therefore, we used ovaries to test the fTRG library and probed the expression of 115 randomly selected lines in germline cells versus somatic cells during oogenesis (*Figure 3A*). From the 115 lines, 91 (79% ) showed detectable expression during oogenesis, with 45 lines being expressed in both, germ cells and the somatic epithelial cells (*Figure 3B,C* and *Supplementary file 3*). 76 (66% ) fTRG lines showed interesting expression patterns restricted to subsets of cells or to a subcellular compartment (*Figure 3B-D*). For example, Tan-GFP is expressed in germline stem cells only, whereas the ECM protein

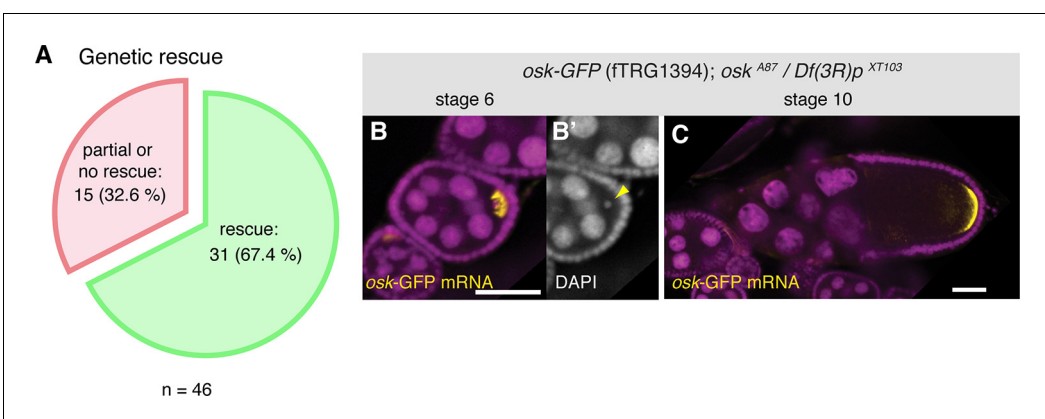

**Figure 2.** Functionality tests of the GFP-tagged fTRG lines by genetic complementation analysis. (A) Genetic rescue statistics of null/strong mutant alleles for 46 selected fTRG lines. Note that more than two-thirds of the lines show a rescue (see *Table 2*). (B, C) *osk*-GFP mRNA (in yellow) expressed from fTRG1394 rescues egg-chamber development of an *osk* null allele (*Jenny et al., 2006*). *osk*-GFP mRNA enriches in the early oocyte (B, stage 6) and rescues the oogenesis arrest and the DNA condensation defect of the *osk* mutant (B', yellow arrowhead). At stage 10 *osk*-GFP RNA enriches at the posterior pole (C) and produces sufficient protein to ensure proper embryogenesis. *osk*-GFP mRNA is shown in yellow, DAPI in magenta; scale bars indicate 30 μm.

**Table 2.** Genetic rescue of mutants with the fTRG lines Table listing fTRG lines and respective genetic alleles as well as rescue assays that were used to assess the functionality of the fTRG lines. Note that about two-thirds of the lines rescue the mutant phenotypes.

| Gene | Chromosome | fTRG line | Tag | Rescue? | Rescue assay | Alleles, deficiencies used in trans for rescue assay | Reference |
|---|---|---|---|---|---|---|---|
| amos | 2nd | fTRG_218 | 2xTY1-sGFP-V5-preTEV-BLRP-3xFLAG | yes | antenna size and bristle number rescued to normal | amos[3] | |
| anterior open (aop, Yan) | 2nd | fTRG_142 | 2xTY1-sGFP-V5-preTEV-BLRP-3xFLAG | yes | embryonic lethality rescued to viable adults | aop[1] (BL-3101); aop[Yan1] (BL-8780) | |
| aubergine (aub) | 2nd | fTRG_581 | 2xTY1-sGFP-V5-preTEV-BLRP-3xFLAG | yes | female sterility entirely rescued | aub[HN2] (BL-8517); Df(2L)BSC145 (BL-9505) | |
| baboon (babo) | 2nd | fTRG_444 | 2xTY1-sGFP-V5-preTEV-BLRP-3xFLAG | yes | lethality rescued to viable adults | babo[32] (BL-5399); babo[k16912] (BL-11207) | |
| bag of marbles (bam) | 3rd | fTRG_3 | 2xTY1-sGFP-V5-preTEV-BLRP-3xFLAG | yes | female sterility entirely rescued | bam[delta86]; Df(3R)exel9020 | Christian Bökel, pers. comm. |
| cactus (cact) | 2nd | fTRG_516 | 2xTY1-sGFP-V5-preTEV-BLRP-3xFLAG | yes | lethality and female sterility rescued | cact[1]; cact[4] | |
| CG32121 | 3rd | fTRG_92 | 2xTY1-sGFP-V5-preTEV-BLRP-3xFLAG | yes | flightlessness rescued | CG32121 [XZ1] (X. Zhang and F.S., unpublished); Df(3L)ED4502 (BL-8097) | |
| CG6509 (dlg5) | 2nd | fTRG_10251 | 2xTY1-sGFP-3xFLAG | yes | lethality rescued to viable adults | CG6509 [KG006748] (BL13692); Df(2L)BSC244 (BL-9718) | |
| discs large 1 (dlg1) | X | fTRG_502 | 2xTY1-sGFP-V5-preTEV-BLRP-3xFLAG | yes | male lethality rescued to viable adults | Dlg1[5] (BL-36280) | |
| dorsal (dl) | 2nd | fTRG_29 | 2xTY1-sGFP-V5-preTEV-BLRP-3xFLAG | yes | bristle number rescued to normal | dl[1]; dl[4] | |
| ebi | 2nd | fTRG_10141 | 2xTY1-sGFP-3xFLAG | yes | lethality rescued to viable adults | ebi[CCS-8] (BL-8397); ebi[E90] (BL-30720) | |
| escargot (esg) | 2nd | fTRG_10170 | 2xTY1-sGFP-3xFLAG | no | lethality not rescued | esg[35Ce-1] (BL-3900); esg[35Ce-3] (BL-30475) | |
| eyes absent (eya) | 2nd | fTRG_492 | 2xTY1-sGFP-V5-preTEV-BLRP-3xFLAG | no | lethality not rescued | eya[CO233]; eya[CO275] | |
| fat (ft) | 2nd | fTRG_10233 | 2xTY1-T2A-nlsGFP-3xFLAG | yes | lethality rescued to viable adults | ft[G-rv] (BL-1894); ft[8] (BL-5406) | |
| 48 related 2 (Fer2) | 3rd | fTRG_334 | 2xTY1-sGFP-V5-preTEV-BLRP-3XFLAG | yes | defective climbing rescued to wild type | Fer2[e03248] | *Dib et al., 2014* |
| fizzy (fzy) | 2nd | fTRG_10250 | 2xTY1-T2A-nlsGFP-3xFLAG | no | lethality not rescued | fzy[1] (BL-2492); fzy[3] (BL-25143) | |

*Table 2 continued on next page*

Table 2 continued

| Gene | Chromosome | fTRG line | Tag | Rescue? | Rescue assay | Alleles, deficiencies used in trans for rescue assay | Reference |
|---|---|---|---|---|---|---|---|
| flightless I (fliI) | X | fTRG_467 | 2xTY1-sGFP-V5-preTEV-BLRP-3xFLAG | yes | lethality or flightlessness rescued | fliI[14] (BL-7481); fliI[3] (BL-4730) | |
| Hand | 2nd | fTRG_10163 | 2xTY1-sGFP-3xFLAG | yes | semi-lethality rescued to viable adults | Hand[173] | |
| hippo (hpo) | 2nd | fTRG_10130 | 2xTY1-sGFP-3xFLAG | yes | larval lethality rescued to viable adults | hpo[KS240] (BL-25085); hpo[KC202] (BL-25090) | |
| HLH54F | 2nd | fTRG_153 | 2xTY1-sGFP-V5-preTEV-BLRP-3xFLAG | yes | lethality rescued to viable adults | bHLH54F[598]; Df(2R) Exel7150 (BL-7891) | |
| Integrin linked kinase (Ilk) | 3rd | fTRG_483 | 2xTY1-sGFP-V5-preTEV-BLRP-3xFLAG | yes | embryonic lethality rescued to viable adults (wing blisters) | Ilk[1] (BL-4861); Df(3L)BSC733 (BL-26831) | |
| Kinesin heavy chain (Khc) | 2nd | fTRG_10243 | 2xTY1-T2A-nlsGFP-3xFLAG | yes | lethality rescued to viable adults | Khc[8] (BL-1607); Khc[1ts] (BL-31994) | |
| LanB1 | 2nd | fTRG_681 | 2xTY1-sGFP-V5-preTEV-BLRP-3xFLAG | yes | lethality rescued to viable adults | LanB1 [KG03456] (BL-13957); Df(2L) Exel7032 (BL-7806) | |
| multiple ankyrin repeats single KH domain (mask) | 3rd | fTRG_486 | 2xTY1-sGFP-V5-preTEV-BLRP-3xFLAG | yes | lethality rescued to viable adults | mask[10.22]/ Df(3R)BSC317 | Barry Thompson, pers. comm |
| midline (mid) | 2nd | fTRG_490 | 2xTY1-sGFP-V5-preTEV-BLRP-3xFLAG | no | lethality not rescued | mid[B1295]; mid[C2372] | |
| numb | 2nd | fTRG_25 | 2xTY1-sGFP-V5-preTEV-BLRP-3xFLAG | yes | lethality rescued to viable adults | numb[1] (BL-4096); Df(2L)30A-C (BL-3702) | |
| odd skipped (odd) | 2nd | fTRG_47 | 2xTY1-sGFP-V5-preTEV-BLRP-3xFLAG | no | lethality not rescued | odd[5] (BL-5345); Df(2L) Exel7018 (BL-7789) | |
| optomotor-blind-related-gene-1 (org-1) | X | fTRG_485 | 2xTY1-sGFP-V5-preTEV-BLRP-3xFLAG | no | male lethality not rescued | org-1[OJ487] | |
| oskar (osk) | 3rd | fTRG_1394 | 2XTY1-SGFP-V5-preTEV-BLRP-3xFLAG | yes | female sterility entirely rescued | osk[A87]/ Df(3R)p-XT103 | |
| Pabp2 | 2nd | fTRG_565 | 2xTY1-sGFP-V5-preTEV-BLRP-3xFLAG | no | lethality not rescued | Pabp2[01] (BL-9838); Pabp2[55] (BL-38390) | |
| patched (ptc) | 2nd | fTRG_82 | 2xTY1-sGFP-V5-preTEV-BLRP-3xFLAG | yes | lethality rescued to viable adults | ptc[9] (BL-3377); ptc[16] (BL-35500) | |
| retina abarrent in pattern (rap) | X | fTRG_1253 | 2xTY1-sGFP-V5-preTEV-BLRP-3xFLAG | yes | lethality rescued to viable adults | rap[ie28] | Yuu Kimata, pers. comm. |

Table 2 continued on next page

*Table 2 continued*

| Gene | Chromosome | fTRG line | Tag | Rescue? | Rescue assay | Alleles, deficiencies used in trans for rescue assay | Reference |
|---|---|---|---|---|---|---|---|
| rhea (Talin) | 3rd | fTRG_587 | 2xTY1-sGFP-V5-preTEV-BLRP-3XFLAG | yes | embryonic lethality rescued to viable adults | rhea[1]; rhea[79] | Jörg Großhans, pers. comm. |
| RhoGEF2 | 2nd | fTRG_591 | 2xTY1-sGFP-V5-preTEV-BLRP-3XFLAG | yes | embryonic lethality rescued to viable adults | RhoGEF2 [04291] | |
| roundabout (robo) | 2nd | fTRG_567 | 2xTY1-sGFP-V5-preTEV-BLRP-3XFLAG | no | lethality not rescued | robo[1] (BL-8755); robo[2] (BL-8756) | |
| saxophone (sax) | 2nd | fTRG_10070 | 2xTY1-sGFP-3xFLAG | yes | lethality rescued to viable adults | sax[4] (BL-5404); sax[5] (BL-8785) | |
| scribbler (sbb) | 2nd | fTRG_443 | 2xTY1-sGFP-V5-preTEV-BLRP-3xFLAG | no | lethality not rescued | sbb[04440] (BL-11376); Df(2R)BSC334 (BL-24358) | |
| Sin3A | 2nd | fTRG_596 | 2xTY1-sGFP-V5-preTEV-BLRP-3XFLAG | no | lethality not rescued | Sin3A[08269] (BL-12350); Sin3A [B0948] | |
| smoothened (smo) | 2nd | fTRG_599 | 2xTY1-sGFP-V5-preTEV-BLRP-3XFLAG | yes | lethality rescued to viable adults | smo[3] (BL-3277); smo[119B6] (BL-24772) | |
| snail (sna) | 2nd | fTRG_71 | 2xTY1-sGFP-V5-preTEV-BLRP-3xFLAG | no | lethality not rescued | sna[18] (BL-2311); sna[1] (BL-25127) | |
| spalt major (salm) | 2nd | fTRG_165 | 2xTY1-sGFP-V5-preTEV-BLRP-3xFLAG | no | lethality not rescued | salm[1] (BL-3274); Df(2L)32FP-5 (BL-29717) | |
| Target of rapamycin (Tor) | 2nd | fTRG_713 | 2xTY1-sGFP-V5-preTEV-BLRP-3XFLAG | no | lethality not rescued | Tor[deltaP] (BL-7014); Df(2L) Exel7055 (BL-7823) | |
| traffic jam (tj) | 2nd | fTRG_163 | 2xTY1-sGFP-V5-preTEV-BLRP-3xFLAG | no | sterility not rescued | tj[PL3] (BL-4987); Df(2L)Exel8041 (BL-7849) | |
| viking (vkg) | 2nd | fTRG_595 | 2xTY1-sGFP-V5-preTEV-BLRP-3XFLAG | no | lethality not rescued | vkg[01209] (BL-11003); Df(2L) Exel7022 (BL-7794) | |
| Unc-89/ Obscurin | 2nd | fTRG_1046 | 2xTY1-sGFP-V5-preTEV-BLRP-3XFLAG | yes | flightlessness rescued | Unc-89[EY15484] | |
| yorkie (yki) | 2nd | fTRG_875 | 2xTY1-sGFP-V5-preTEV-BLRP-3XFLAG | yes | lethality rescued to viable adults | yki[B5] | Barry Thompson, pers. comm. |

Pericardin (Prc-GFP) is concentrated around the neighbouring cap cells, and the transcription factor Delilah (Dei-GFP) is specifically localised to the nuclei of somatic stem cells, which will give rise to the epithelial cells surrounding each egg chamber (*Figure 3A,C*). In early egg chambers, Reph-GFP is expressed in germ cells only, whereas the ECM protein Viking (Vkg-GFP) specifically surrounds all the somatic epithelial cells. Interestingly, the transcription factor Auracan (Ara-GFP) is only expressed in posterior follicle cells, whereas the putative retinal transporter CG5958 is only detectable in the squamous epithelial cells surrounding the nurse cells (*Figure 3C*).

We further investigated the subcellular localisation of the tagged proteins, which revealed a localisation for the RNA helicase l(2)35Df to all nuclei, whereas the predicted $C_2H_2$-Zn-finger transcription factor Crooked legs (Crol-GFP) is restricted to the nuclei of the epithelial cells (*Figure 3D*). Interestingly, Corolla-GFP is exclusively localised to the oocyte nucleus in early egg chambers. This is consistent with the function of Corolla at the synaptonemal complex attaching homologous chromosomes during early meiosis (*Collins et al., 2014*). In contrast, the uncharacterised homeobox transcription factor E5 (E5-GFP) is largely restricted to the nuclei of anterior and posterior epithelial cells (*Figure 3D*). Apart from nuclear patterns, we found a significant number of cortical localisations, including the well characterised Crumbs (Crb-GFP) (*Bulgakova and Knust, 2009*) and the PDZ-domain containing Big bang (Bbg-GFP) (*Bonnay et al., 2013*) at the apical cortex of the epithelial cells, the $Na^+/K^+$ transporter subunit Nervana 2 (Nrv2-GFP) at the lateral epithelial membrane, and the EGF-signalling regulator Star (S-GFP) as well as the TGF-β receptor Saxophone (Sax-GFP) localised to the cortex or membrane of the germ cells (*Figure 3D* and *Supplementary file 3*). Furthermore, we find a perinuclear enrichment for the uncharacterised predicted NAD-binding protein CG8768, and oocyte enrichments for the Tom22 homolog Maggie (Mge-GFP) (*Vaskova et al., 2000*), the glycosyltransferase Wollknäuel (Wol-GFP) (*Haecker et al., 2008*) and the TGF-α homolog Gurken (Grk-GFP), the latter with its well-established concentration around the oocyte nucleus (*Neuman-Silberberg and Schüpbach, 1993*), where it co-localises with endogenous Grk protein (*Figure 3D* and *Figure 3—figure supplement 1A,B*).

We did not perform genetic rescue assays for all of lines examined for protein localisation. However, even tagged proteins, for which the rescue assay failed, such as Vkg-GFP, can result in an informative expression pattern (*Figure 3D*, *Table 2*). Obviously, an independent validation of expression patterns established solely based on a tagged transgenic reporters is advisable.

To test if genes expressed from the FlyFos system also undergo normal post-transcriptional regulation, we analysed the *osk-GFP* line, which was recently used as a label for germ granules (*Trcek et al., 2015*). *osk* mRNA is transcribed from the early stages of oogenesis onwards in the nurse cell nuclei and specifically transported to the oocyte, where it localises to the posterior pole (*Ephrussi et al., 1991*; *Kim-Ha et al., 1991*). Only after the mRNA is localised, it is translated from stage 9 onwards (*Kim-Ha et al., 1995*). Indeed, fosmid derived *osk-GFP* mRNA localises normally during all stages of oogenesis and its translation is repressed during mRNA transport, as Osk-GFP protein can only be detected at the posterior pole from stage 9 onwards (*Figure 3—figure supplement 2A,B*). *osk*-GFP also rescues all aspects of an *osk* null allele (*Figure 2B,C*) and Osk-GFP colocalises with endogenous Osk protein (*Figure 3—figure supplement 1C,D*). Additionally, we discovered a post-transcriptional regulation for *corolla. corolla-GFP* mRNA localises to the oocyte at stage 6 and Corolla-GFP protein is transported into the oocyte nucleus. However, despite the presence of the *corolla-GFP* mRNA at stage 8, Corolla protein is undetectable, suggesting either a translational block of the RNA or targeted degradation of the protein (*Figure 3—figure supplement 2C-F*). Taken together, these expression and protein localisation data recapitulate known patterns accurately and identify various unknown protein localisations in various cell types during oogenesis, and thus emphasise the value of the fly TransgeneOme resource.

## Live in toto imaging of fTRG lines during embryogenesis

For many genes, the expression patterns at the mRNA level are particularly well characterised during *Drosophila* embryogenesis (*Hammonds et al., 2013*; *Tomancak et al., 2002*; *2007*). However, in situ hybridisation techniques on fixed tissues do not visualise the dynamics of expression over time and thus do not allow tracking of the expressing cells during development. As our tagging approach enables live imaging at endogenous expression levels, we set out to test if *in toto* imaging using the SPIM (Selective Plane Illumination Microscopy) technology (*Huisken et al., 2004*) can be applied to the fly TransgeneOme lines. We pre-screened a small subset of lines (*Table 3*) and selected the $Na^+/$

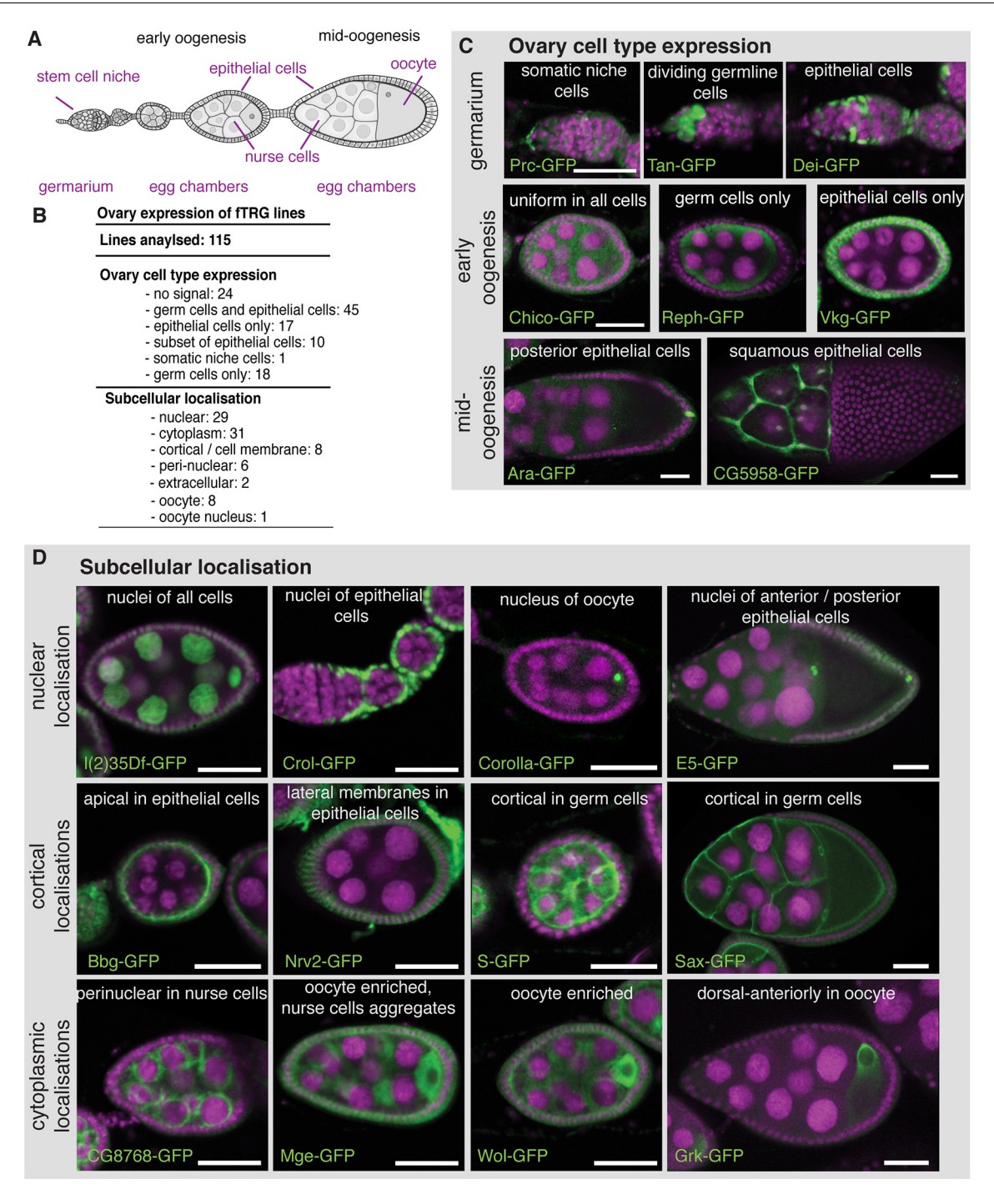

**Figure 3.** Expression of fTRG tagged proteins in ovaries. (**A**) Schematic overview of oogenesis stages and cell types. (**B**) Summary of the identified expression patterns; see also **Supplementary file 3**. (**C**) Selected examples for cell type specific fTRG expression patterns at germarium, early- and mid-oogenesis stages visualised by anti-GFP antibody staining. (**D**) Selected examples of subcellular localisation patterns, highlighting nuclear, cortical and cytoplasmic patterns at different oogenesis stages. GFP is show in green, DAPI in magenta; scale bars indicate 30 μm.

The following figure supplements are available for figure 3:

**Figure supplement 1.** Co-localisation of fTRG derived tagged proteins with endogenous proteins during oogenesis.

**Figure supplement 2.** Posttranscriptional regulation of protein expression during oogenesis.

K$^+$ transporter subunit Nrv2, as it shows high expression levels, for long-term time-lapse live imaging with a multi-view dual-side SPIM (*Huisken and Stainier, 2007*). During embryogenesis Nrv2 expression was reported in neurons (*Sun et al., 1999*) and glial cells (*Younossi-Hartenstein et al., 2002*). Interestingly, we find that Nrv2-GFP is already expressed from stage 11 onwards in most likely all cell types, where it localises to the plasma membrane (*Figure 4A-C*), similarly to its localisation in ovaries (*Figure 3D*). The expression level increases during stage 15 in all cells, with a particularly strong increase in the developing central nervous system (CNS) labelling the CNS and motor neuron membranes (*Figure 4*, *Video 1*, Examine raw data in BigDataViewer (Fiji -> Plugins -> BigData-Viewer -> Browse BigDataServer and enter http: //bds.mpi-cbg.de:8087). These live *in toto* expression data are consistent with expression data of a recently isolated GFP trap in *nrv2* (*Lowe et al., 2014*), thus validating our methodology.

We wanted to extend our approach beyond highly expressed structural genes towards transcription factors that enable to follow cell lineages in the embryo. For this purpose, we crossed the fTRG line of the homeobox transcription factor *gooseberry* (Gsb-GFP) to H2A-mRFPruby, which labels all nuclei, and recorded a two-colour multi-view dual-side SPIM Video. We find that Gsb-GFP is expressed in the presumptive neuroectoderm of the head region, labelling segmentally reiterated stripe-like domains at stage 10 (*Figure 4—figure supplement 1A,B*, *Video 2*) as was described from fixed images (*Gutjahr et al., 1993*). Focusing on the deuterocerebral domain, we could reconstruct the delamination of two neuroblasts, which up-regulate Gsb-GFP while initiating their asymmetric divisions (*Figure 4—figure supplement 1C-F*). It was possible to individually follow their neural progeny. Gsb-GFP expression also allowed us to directly follow the gradual down-regulation of Gsb-GFP in ectodermal cells that remained at the head surface after neuroblast delamination. As opposed to the neuroblasts, these cells, which give rise to epidermis, did not divide at all, or underwent only one further division (*Figure 4—figure supplement 1G-J*, *Video 2*).

*gsb* is in part required for *gooseberry-neuro* (*gsb-n*; also called *gsb-d*) expression (*He and Noll, 2013*). Notably, the expression of Gsb-n-GFP (fTRG line 513) becomes detectable only at the end of germ-band extension (end of stage 11) in the developing CNS, where it lasts until stage 17 (*Figure 4—figure supplement 2*, *Video 3*). Again, this is consistent with published immuno-histochemistry data (*Gutjahr et al., 1993*; *He and Noll, 2013*). We conclude that our fly TransgeneOme library can be used for live *in toto* imaging, even for transcription factors expressed at endogenous levels. This will be of significant importance for on-going efforts linking the transcription factor expression patterns of embryonic neuroblasts to the morphologically defined lineages that structure the larval and adult *Drosophila* brain (*Hartenstein et al., 2015*; *Lovick et al., 2013*; *Pereanu, 2006*).

## Expression of fTRG lines in the adult thorax

Cells and tissues in the embryo are not yet terminally differentiated. To apply our TransgeneOme library to fully differentiated tissues, we decided to study tissues of the adult thorax. We scored expression in the large indirect flight muscles (IFMs), in leg muscles, in the visceral muscles

**Table 3.** in totoSPIM imaging of fTRG lines in the embryo Table listing the fTRG lines that were imaged in the embryo using Zeiss Lightsheet Z.1 from multiple angles over time. *nrv2*, *gsb* and *gsb-n* are discussed in the text. For the remaining lines, we list broad categorisation of the expression detected by SPIM imaging.

| fTRG number | FBgn_id | Gene symbol | Signal | Embryonic expression | Movie | Beads |
|---|---|---|---|---|---|---|
| 58 | FBgn0001148 | *gsb* | strong | tissue-specific expression | Yes | Yes |
| 71 | FBgn0003448 | *snail* | weak | tissue-specific expression | Yes | Yes |
| 88 | FBgn0025360 | *Optix* | medium | tissue-specific expression | Yes | Yes |
| 94 | FBgn0010433 | *ato* | weak | tissue-specific expression | Yes | Yes |
| 137 | FBgn0259685 | *crb* | medium | tissue-specific expression | Yes | Yes |
| 155 | FBgn0029123 | *SoxN* | strong | tissue-specific expression | Yes | Yes |
| 349 | FBgn0024294 | *spn43Aa* | strong | late expression, deposited in the cuticle | Yes | Yes |
| 513 | FBgn0001147 | *gsb-n* | medium | tissue-specific expression | Yes | Yes |
| 937 | FBgn0015777 | *nrv2* | strong | ubiquitous expression, membrane signal | Yes | Yes |

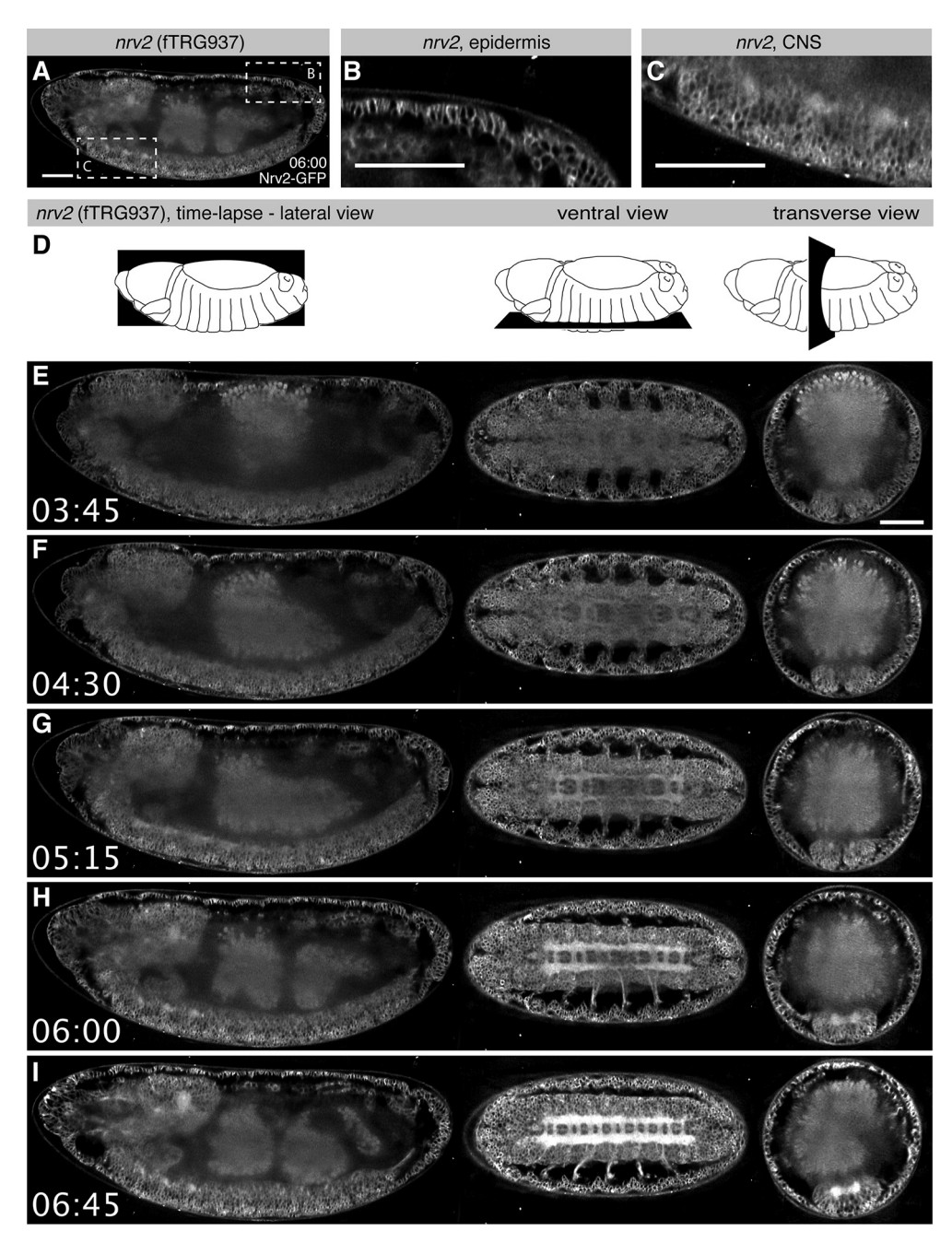

**Figure 4.** Live in toto imaging during embryogenesis with SPIM. (A-C) Nrv2-GFP protein is enriched in cell membrane of the epidermis and the CNS of late stage 16 embryos, as shown by a lateral section (A) and high magnifications of the posterior epidermis (B) and the ventral CNS (C). (D) Schemes of the lateral, ventral and transverse optical section views through the embryo shown in (E-I). (E-I) Still image from a Nrv2-GFP time-lapse Video with lateral section views on the left, ventral sections in the middle and transverse sections on the right. Note that Nrv2-GFP is first expressed in the developing epidermal epithelial cells (E, F) and then becomes enriched in the CNS (G-I, see *Video 1*). Scale bars indicate 50 µm.

The following figure supplements are available for figure 4:

**Figure supplement 1.** Live Gsb-GFP imaging during embryogenesis with SPIM.

**Figure supplement 2.** Live Gsb-n-GFP imaging during embryogenesis with SPIM.

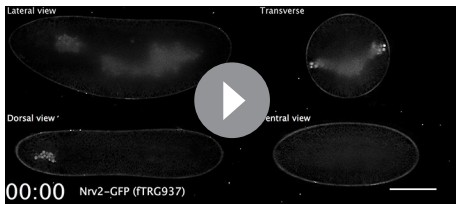

**Video 1.** Multi-view SPIM Video of a stage 12 Nrv2-GFP expressing embryo. A stack was acquired every 15 min, lateral, dorsal, ventral and transverse views of the same time points are displayed. From stage 11 onwards Nrv2-GFP is present ubiquitously in the plasma membrane. Later, its expression increases in the CNS, particularly in the neuropil and the motor neurons. Video plays with 7 frames per second. Time is given in hh:mm. Scale bar indicates 50 µm.

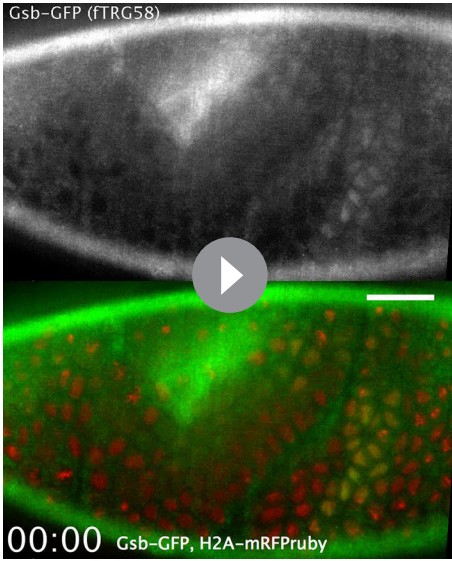

**Video 2.** Lateral head section from a SPIM Video of a stage 10 Gsb-GFP (green, white in the top Video), Histone-2A-mRFPruby (red) embryo. A stack was acquired every 7 min. The segmentally re-iterated stripe-like *gsb* expression domain in the head neuroectoderm is visible. Later, *gsb* is expressed in ganglion mother cells and nerve cells that are the progeny of *gsb* expressing neuroblasts. Video plays with 7 frames per second. Time is given in hh:mm. Scale bar indicates 50 µm.

surrounding the gut, in the gut epithelium, the tendon epithelium, the trachea and the ventral nerve cord including the motor neurons. In total, we found detectable expression in at least one tissue for 101 of 121 (83.5% ) analysed fTRG lines, thus creating a large number of valuable markers for cell types and subcellular structures (*Table 4*, *Supplementary file 4*).

The large IFMs are fibrillar muscles, which have a distinct transcriptional program resulting in their distinct morphology (*Schönbauer et al., 2011*). This is recapitulated by the expression of Act88F-GFP, which localises to the thin filaments of IFMs only (*Figure 5A-C*), whereas Mlp84B-GFP is not expressed in IFMs but at the peripheral Z-discs of leg and visceral muscles only (*Figure 5D-F*), similar to the published localisation in larval muscle (*Clark et al., 2007*). We find various dotty patterns indicating localisation to intracellular vesicles; a particularly prominent example is Tango1-GFP in the midgut epithelium (*Figure 5G,H*, *Supplementary file 4*). Tango1 regulates protein secretion in S2 cells, where it localises to the Golgi apparatus upon over-expression (*Bard et al., 2006*), suggesting that the pattern described here is correct. We find Par6 with an in analogy to other epithelia expected apical localisation (*Hutterer et al., 2004*) in the epithelium of the proventriculus and in trachea (*Figure 5I,J*), whereas we identified a surprising pattern for the TRP channel Painless (*Tracey et al., 2003*). Pain-GFP is not only highly expressed in motor neurons (*Figure 5K*) but also in a particular set of small cells within the gut epithelium and most surprisingly, in the tendon cells to which the IFMs attach (*Figure 5L,M*). To clarify the identity of the Pain-GFP positive cell type in the gut, we co-stained with the differentiated enteroendocrine cell marker Prospero (*Ohlstein and Spradling, 2005*), however, did not find any overlap with the Pain-GFP positive cells (*Figure 5Q*). This suggests that the small, likely diploid, Pain-GFP positive cells are either enteroblasts or intestinal stem cells (ISCs), the source of all enterocytes and enteroendocrine cells that build the gut

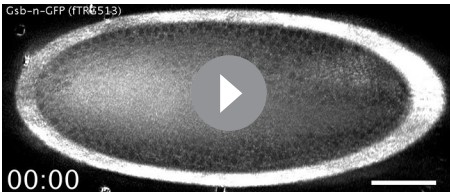

**Video 3.** Ventral view of a SPIM Video of a stage 6 Gsb-n-GFP embryo. A stack was acquired every 15 min. Gsb-n-GFP is only detectable at the end of germ-band extension. During germ-band retraction, it is expressed in characteristic L-shaped expression domains in the hemi-segments of the trunk. In the late stage embryo, Gsb-n-GFP is present in the neurons of the shortening ventral nerve cord. Video plays with 7 frames per second. Time is given in hh:mm. Scale bar indicates 50 µm.

**Table 4.** Summary of adult muscle fTRG expression patterns 54 detected adult muscle localisation patterns (flight muscle, leg muscle and visceral muscle) from **Supplementary file 4** are summarised. fTRG line number is listed in brackets.

| Thick filament | Thin filament / myofibril | M-line | Z-disc | Muscle attachment site | T-tubules / sarcolemma | Dotty pattern / vesicles (?) | Mitochondria | Nucleus | Neuro-muscular junction |
|---|---|---|---|---|---|---|---|---|---|
| Fln (876, IFM) | Act88F (78, 10028, IFM) | Prm (475, IFM) | CG31772 (63) | Ilk (483) | Dlg1(502) | Babo (444) | CG12118 (276) | Adar (570) | Cact (516) |
| Mf (Iso-A,G, N, 501) | Fray (125, 10032) | Unc-89 (1046) | Kettin (Sls-Isoform, 569) | Talin (*rhea*, Iso-B, E, F, G, 587) | Sax (10070) | CG5885 (10017, leg m.) | | Blimp-1 (10149) | Veli (10125) |
| Mhc (Iso-K, L, M, 500) | Hsp83 (10010) | | Lmpt (584, I-band, leg m.) | β-PS Integrin (*mys*, 932) | | CLIP-190 (156) | | CG11617 (10041) | |
| Mhc (Iso-A, F, 519, leg m. subset & visceral. m.) | TpnC25D (1257, leg m. & visceral m.) | | Mask (486, IFM) | | | Dlg5 (*CG6509*, 10251, IFM) | | CG12391 (10036) | |
| Prm (475, leg m. & visceral m.) | TpnI (*wupA*, 925, leg m. & visceral m.) | | Mlp60A (709, leg m. & visceral m.) | | | Hts (585) | | CG17912 (10035) | |
| | | | Mlp84B (678, leg m. & visceral m.) | | | Mask (486, leg m.) | | CG32121 (92) | |
| | | | Talin (Iso-C, D, 731, leg m. & visceral m.) | | | Pyd3 (53) | | Dorsal (29, leg m.) | |
| | | | | | | Rho1 (31) | | E2F2 (115) | |
| | | | | | | Sc2 (79, 10039) | | Gro (21) | |
| | | | | | | Tango11 (699) | | Hand (10163, visceral m.) | |
| | | | | | | Tsc1 (59) | | Hb (139, leg m.) | |
| | | | | | | Vps35 (*CG5625*, 57) | | Mnt (34) | |
| | | | | | | | | P53 (84) | |
| | | | | | | | | Salm (165) | |
| | | | | | | | | Vri (182) | |

epithelium (*Jiang and Edgar, 2011*). At this point, we can only speculate that Pain might be involved in mechanical stretch-sensing in these cell types. We have also tagged various ECM components, with LamininB1 (LanB1-GFP), LamininA (LanA-GFP) and BM40-SPARC resulting in the most prominent expression patterns. All three ensheath most adult tissues, particularly the muscles (*Figure 5N, O*, *Figure 5—figure supplement 1A,B,E,F*). Interestingly, LanA-GFP and LanB1-GFP also surround the fine tracheal branches that penetrate into the IFMs, whereas BM40-SPARC is only detected around the large tracheal stalk and the motor neurons (*Figure 5P*, *Figure 5—figure supplement 1C,D,G,H*). Finally, we also detected prominent neuromuscular junction (NMJ) markers; the IκB homolog Cactus shows a distinct pattern on leg muscles, visceral muscles and IFMs, the latter we could confirm by co-staining with the neuronal marker Futsch (*Figure 5—figure supplement 1I-M*). Interestingly, such a NMJ pattern for Cactus and its binding partner Dorsal has been shown in larval body muscle by antibody stainings (*Bolatto et al., 2003*). Together, these results suggest that our

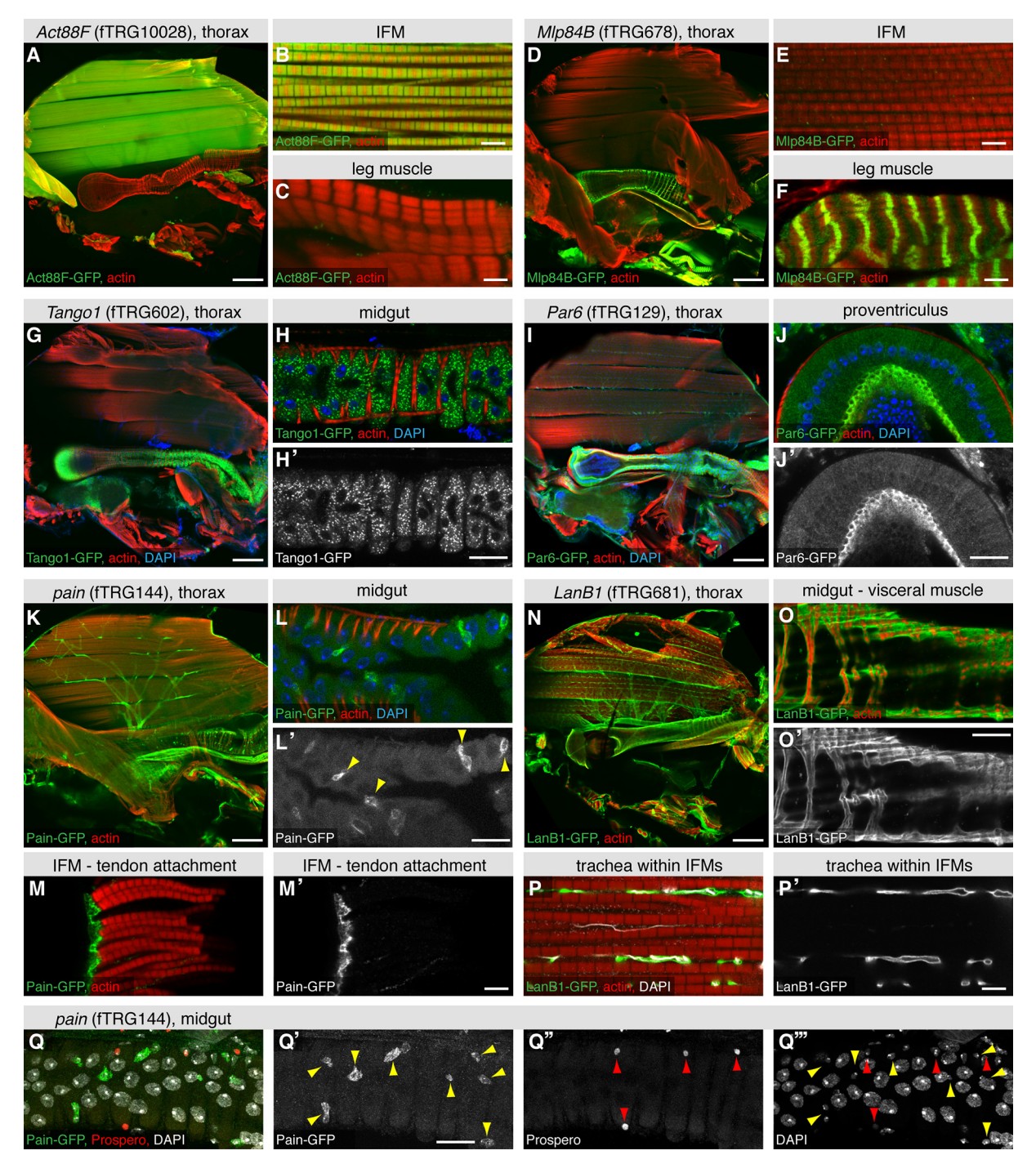

**Figure 5.** Expression of fTRG tagged proteins in tissues of the adult thorax. Antibody stainings of the adult thorax with anti-GFP antibody (green) and phalloidin (red). (**A-F**) Act88-GFP expression is specific to the IFMs, where it labels the thin filaments (**B**), whereas Mlp84B specifically labels the Z-discs of leg muscles (**F**). (**G-J**) Tango1-GFP concentrates in a vesicle-like pattern in the gut epithelium (**H, H'**), whereas Par6-GFP is highly expressed in trachea (**I**) and the gut epithelium, where it concentrates at the apical membrane, as shown for a cross-section of the proventriculus (**J, J'**), nuclei are labelled with DAPI (blue). (**K-M**) Pain-GFP expression in the flight muscle motor neurons (**K**), small cells within the midgut epithelium (**L, L'**) and tendon cells (**M, M'**). (**N-P**) LanB1-GFP labels the extracellular matrix surrounding the IFMs, the motor neurons and the trachea (**N**), as well as the visceral muscles (**O**). Even the finest trachea marked by UV auto-fluorescence (white) (**P**) are surrounded by LanB1-GFP (**P'**). (**Q**) Pain-GFP positive cells in the midgut do not overlap with Prospero positive nuclei of enterocytes (in red) and contain small nuclei, as visualised by DAPI co-stain in white (**Q-Q'''**). Scale bars indicate 100 µm (**A, D, I, K, N**), 20 µm (**H,J, L, O, Q**) and 5 µm (**B,C,E,F,M,P**).

*Figure 5 continued on next page*

*Figure 5 continued*

The following figure supplement is available for figure 5:

**Figure supplement 1.** Extracellular matrix and synaptic markers of the adult thorax.

fly TransgeneOme library provides a rich resource for tissue-specific markers in the adult fly that can routinely be used to visualise subcellular compartments in various tissues.

To further validate the advantages of our TransgeneOme lines to label subcellular structures, we imaged the large IFMs of the same 121 lines at high resolution. We found various markers for the thick filaments (e.g. the myosin-associated protein Flightin, Fln-GFP) (*Vigoreaux et al., 1993*), for the myofibrils (e.g. the protein kinase Fray-GFP), the M-lines (e.g. the titin related protein Unc-89/Obscurin-GFP) (*Katzemich et al., 2012*), the Z-discs (e.g. CG31772-GFP) and the muscle attachment sites (e.g. Integrin-linked-kinase, Ilk-GFP). Furthermore, we identified markers for the T-tubules (e.g. Dlg1-GFP), for different vesicular compartments (e.g. the TGFβ receptor Baboon-GFP) and for mitochondria (CG12118) within the IFMs (*Figure 6*, *Table 4*, *Supplementary file 4*). The latter was confirmed by co-labelling the mitochondria with an antibody against the mitochondrial ATPase (complex V α subunit) (*Cox and Spradling, 2009*) (*Figure 6—figure supplement 1*). Additionally, we documented the nuclear localisation in IFMs and leg muscles for a variety of fTRG proteins, including the uncharacterised homeodomain protein CG11617 and the $C_2H_2$ Zinc-fingers CG12391 and CG17912 (*Figure 6—figure supplement 2A-C,E-G*); both of the latter result in flightless animals when knocked-down by muscle-specific RNAi (*Schnorrer et al., 2010*) suggesting that these genes play an essential role for IFM morphogenesis or function. Interestingly, the well characterised $C_2H_2$ Zinc-finger protein Hunchback (Hb) is only localised to leg muscle nuclei, but absent from IFMs suggesting a leg muscle-specific function of Hb (*Figure 6—figure supplement 2G,H*).

However, differences between muscle types are not only controlled transcriptionally but also by alternative splicing (*Oas et al., 2014*; *Spletter and Schnorrer, 2014*; *Spletter et al., 2015*). To investigate if our tagging approach can be used to generate isoform-specific lines, we have chosen two prominent muscle genes, *mhc* and *rhea* (the fly Talin), both of which have predicted isoforms with different C-termini (*Figure 6—figure supplement 3A,H*). Interestingly, we found that Mhc-isoforms K, L, M are expressed in IFMs and all leg muscles, however the predicted Mhc-isoforms A, F, G, B, S, V with the distal STOP codon are selectively expressed in visceral muscle and in a subset of leg muscles, however absent in adult IFMs (*Figure 6—figure supplement 3B-G*). Even more surprisingly, while the long 'conventional' *rhea* (Talin) isoforms B, E, F, G show the expected localisation to muscle attachment sites in IFMs and leg muscles (*Weitkunat et al., 2014*), the short Talin-isoforms C and D do not localise to muscle attachment sites, but are selectively concentrated at costamers of leg muscles (*Figure 6—figure supplement 3I-N*). Hence, the tagged proteins of our TransgeneOme library are ideally suited to label subcellular compartments and protein complexes, and in some cases they can even distinguish between closely related protein isoforms.

## GFP-tagged proteins largely recapitulate the endogenous protein localisation

Tagging of any protein can in principle affect its localisation pattern in a cell. To investigate the reliability of the observed GFP-tagged protein patterns, we selected eight different fTRG lines with specific GFP patterns in adults described above (LanA, LanB (not shown), Par6, Mlp84B, Mhc, Fln, Unc89/Obscurin and Dlg1), for which reliable antibodies against the corresponding proteins were readily available. In all cases, we observed a high degree of overlap between the pattern revealed by staining with anti-GFP antibody and the respective specific antibody staining pattern in the same transgenic line. This suggests that the tagged proteins expressed in the fTRG lines and the endogenous proteins co-localised to a large extent (*Figure 7A-D,I-L*). To rule out that the tagged protein somehow induces the endogenous protein to adopt an abnormal pattern, we compared the localisation patterns in the fTRG lines (with the protein specific antibody that visualizes both the tagged and untagged endogenous form of the protein) to a wild type strain, which only expresses the endogenous protein (*Figure 7E-H,M-P*). Only in one fTRG line, expressing the highly expressed short Mhc isoforms K, L, M-GFP tagged, we found an abnormally broad anti-Mhc pattern in IFMs, however not

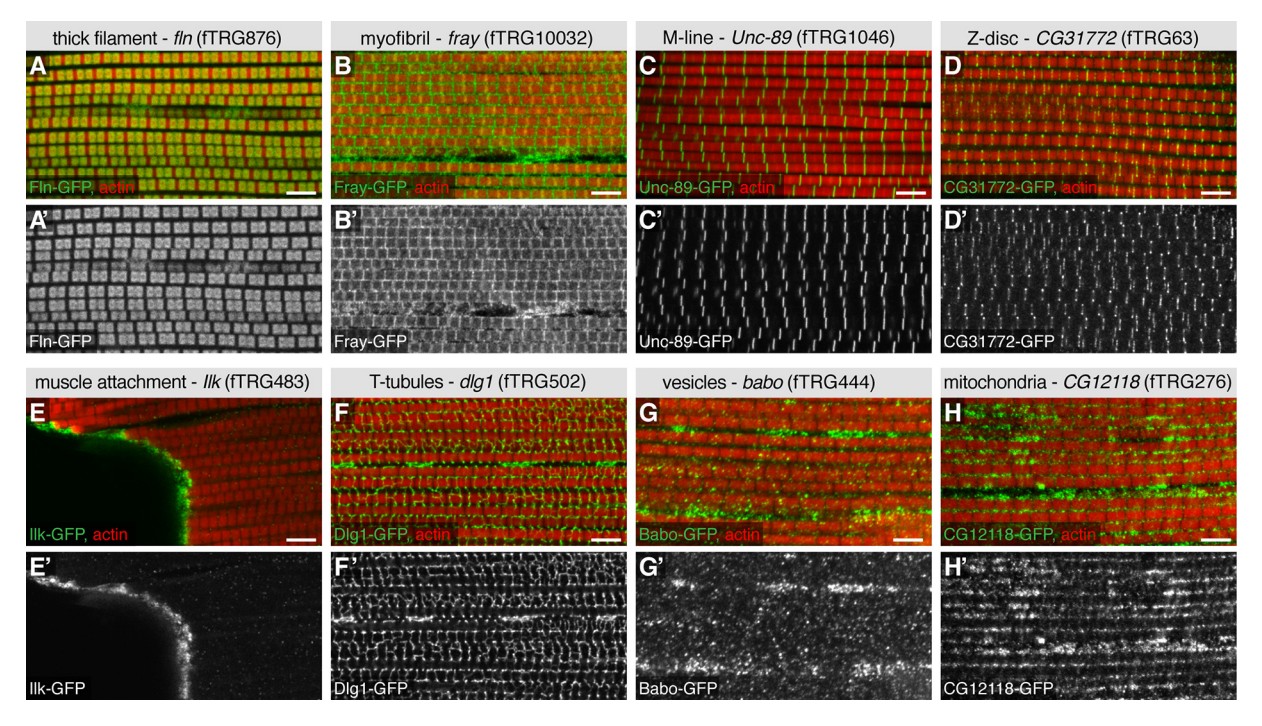

**Figure 6.** Subcellular expression patterns in adult flight muscles. Antibody stainings of the adult thorax with anti-GFP antibody (green or white in the single colour images) and phalloidin (red). (**A-D**) Localisation to specific myofibrillar sub-regions; Fln-GFP marks the thick filaments (**A, A'**), Fray-GFP surrounds the myofibrils with an enrichment at M-lines and Z-discs (**B, B'**), Unc-89-GFP marks only M-lines (**C, C'**) and CG31772-GFP only Z-discs (**D, D'**). (**E-H**) Ilk-GFP strongly concentrates at the muscle-tendon attachment sites (**E, E'**), Dlg1-GFP labels the T-tubular membranes (**F, F'**), Babo-GFP shows a dotty, vesicular pattern (**G, G'**) and CG12118-GFP displays a mitochondrial pattern (**H, H'**). Scale bars indicate 5 µm.

The following figure supplements are available for figure 6:

**Figure supplement 1.** CG12118-GFP localises to mitochondria.

**Figure supplement 2.** Nuclear localisations in adult flight muscles.

**Figure supplement 3.** Alternative splicing into alternative C-termini.

in leg muscles compared to wild type (*Figure 7D,H,L,P*). This pattern is explained by an abnormal myofibril morphology in the IFMs of the fTRG500 line, possibly because of an increased Mhc to actin ratio (4 copies vs. 2 copies), for which IFMs are particularly sensitive (*Cripps et al., 1994*).

Tagging a protein may modify its turn-over rates and thus its expression levels, despite preserving its localisation. To investigate if our tagging approach generally changes protein levels, we chose three different tagged proteins, expressed in different tissues, for which we had functional antibodies and the expected 40 kDa shift caused by the tag should result in a detectable shift compared to the size of the untagged endogenous protein. We made total protein extracts from adults males and run Western blots. When probing with an antibody against the tag, we find the expected sizes for the IFM-specific Fln-GFP, the leg muscle-specific Mlp84B-GFP and the broader expressed Dlg1-GFP, the latter running in several bands due to splice isoforms (*Figure 7Q*). Probing the same extracts with specific antibodies against the respective proteins shows that both Fln-GFP and Dlg1-GFP levels are comparable to the endogenous protein, whereas Mlp84B-GFP is expressed at a lower level (*Figure 7Q*). The latter may be caused by the reduced affinity to the thick filament due to the tag resulting in an unstable protein. Together, these data demonstrate that many of the tagged proteins colocalise with the respective endogenous proteins, as has been observed in other large-scale tagging approaches (*Stadler et al., 2013*), however, exact expression levels can be different from

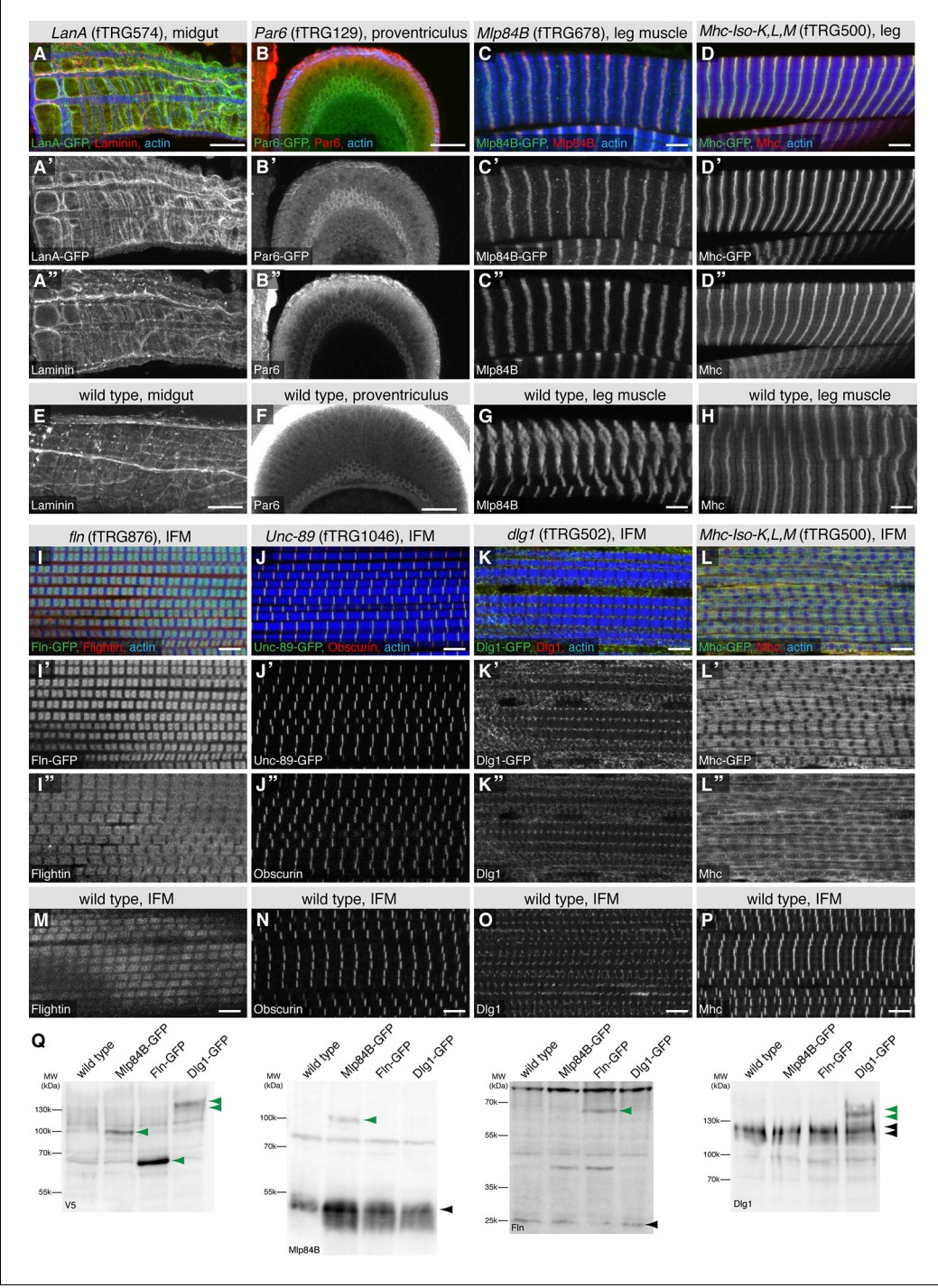

**Figure 7.** Co-localisation of fTRG tagged proteins with endogenous proteins. Antibody stainings of the adult thorax from fTRG or wild-type lines with anti-GFP antibody (green or white in the single colour images) and antibodies against various fly proteins (red). (**A-D**) Co-localisation of LanA-GFP with anti-Laminin antibody stain around the midgut (**A**), of Par6-GFP with anti-Par6 at the apical side of the proventriculus epithelium (**B**) and of Mlp84B-GFP as well as Mhc-GFP with anti-Mlp84B and anti-Mhc antibody stain in leg muscles, respectively (**C, D**). (**E-H**) Adult thoraces from wild-type flies show very similar patterns with the respective antibodies. (**I-L**) Adult IFMs showing the co-localisation of Fln-GFP with anti-Fln antibody staining (**I**), Unc-89/Obscurin-GFP with anti-Obscurin antibody staining (**J**), Dlg1-GFP with anti-Dlg1 antibody staining (**K**) and Mhc-GFP with anti Mhc antibody staining (**L**). (**M-P**) The same antibodies result in very similar patterns in wild-type IFMs apart from the a sharp versus a diffuse Mhc pattern comparing wild-type to Mhc-GFP flies (**L, P**). (**Q**) Western blots loaded with total protein

*Figure 7 continued on next page*

*Figure 7 continued*
extract from wild-type, Mlp84B-GFP, Fln-GFP and Dlg1-GFP adult males probed with anti-V5 (included in the GFP tag) anti-Mlp84B, anti-Fln and anti-Dlg1 antibodies. Note the about 40 kDa size shift of the tagged proteins in the respective lanes (marked with green arrow heads) versus the untagged protein band (black arrow heads).

the wild-type level in some cases. Thus, the fTRG library is a valuable collection of strains to study in vivo protein localisation.

## Expression of the fTRG lines in the living pupal thorax

An attractive application of the fly TransgeneOme library is live in vivo imaging. In the past, we had established live imaging of developing flight muscles in the pupal thorax using over-expressed marker proteins (*Weitkunat et al., 2014*). Here, we wanted to test, if live imaging of proteins at endogenous expression levels is also possible within the thick pupal thorax. We selected six fTRG lines for well established genes and indeed could detect expression and subcellular localisation for all of them using a spinning-disc confocal microscope either at the level of the pupal epidermis or below the epidermis, in the developing flight muscles, or both (*Figure 8*). The adducin-like Hts-GFP labels the cytoplasm of fusing myoblasts from 10 to 20 hr APF (after puparium formation) and the developing SOPs (sensory organ precursors) with a particular prominent concentration in developing neurons and their axons (*Figure 8B-E*). In contrast, Dlg1-GFP localises to cell-cell-junctions of the pupal epidermis and to a network of internal membranes in the developing IFMs (*Figure 8F-I*) that may resemble developing T-Tubules, for which Dlg1 is a well-established marker (*Razzaq et al., 2001*). Interestingly, the long isoforms of Talin-GFP (*rhea* isoforms B, E, F, G) are largely in the cytoplasm and at the cortex of the epidermal cells, with a marked enrichment in the developing SOPs at 10 to 20 hr APF (*Figure 8J,K*). Further, Talin-GFP is strongly concentrated at muscle attachment sites of developing IFMs from 24 hr onwards (*Figure 8L,M*) consistent with antibody stainings of IFMs (*Weitkunat et al., 2014*).

The dynamics of the extracellular matrix is little described thus far as very few live markers existed. Hence, we tested our LamininB1 fosmid and found that LanB1-GFP is readily detectable within the developing basement-membrane basal to the epidermal cells of the pupal thorax at 10 hr APF (*Figure 8N*). It also labels the assembling basement-membrane around the developing IFMs from 16 to 30 hr APF without a particularly obvious concentration at the muscle attachment sites (*Figure 8O-Q*). To specifically visualise the developing IFMs we chose Actin88F, which is specifically expressed in IFMs and a few leg muscles (*Nongthomba et al., 2001*). We find that the Act88F-GFP fTRG line indeed very strongly labels the IFMs from about 18 hr APF but is also expressed in the developing pupal epidermis again with an enrichment in the forming SOPs from 10 to 20 hr APF (*Figure 8R-U*). The latter is not surprising as Act88F-lacZ reporter has been shown to also label the developing wing epithelium (*Nongthomba et al., 2001*), again suggesting that our fTRG line recapitulates the endogenous expression pattern. Finally, we tested the βTub60D fTRG-line, as βTub60D was reported to label the myoblasts and developing myotubes in embryonic and adult muscles (*Fernandes et al., 2005*; *Leiss et al., 1988*; *Schnorrer et al., 2007*). Indeed, we detect βTub60D-GFP in fusing myoblasts and the developing IFMs, with particularly prominent label of the microtubule bundles at 24h APF (*Figure 8V-Y*). In addition, βTub60D-GFP also strongly marks the developing hairs of the sensory organs of the pupal epidermis (*Figure 8X*, see also *Video 6*).

In order to test, if the fly TransgeneOme lines and the sGFP-tag are indeed suited for long-term live imaging in pupae, we chose Act88F-GFP and βTub60D-GFP and imaged the developing IFMs for more than 19 hr with a two-photon microscope using an established protocol for over-expressed markers (*Weitkunat and Schnorrer, 2014*). For both proteins, we can detect strongly increasing expression after 18 hr APF in the developing IFMs, with Act88F-GFP being restricted to the myotubes and the developing myofibrillar bundles (*Video 4*, *Figure 8—figure supplement 1A-F*) whereas βTub60D-GFP also labels the fusing myoblasts and is largely incorporated into prominent microtubule bundles (*Video 6*, *Figure 8—figure supplement 1L-Q*).

As photo bleaching was no serious problem in these long Videos, we also recorded Videos at higher time and spatial resolution. We labelled the developing IFMs with Act88F-GFP and the myoblasts with a *him*-GAL4, UAS-palm-Cherry and acquired a 3D stack every two minutes using a

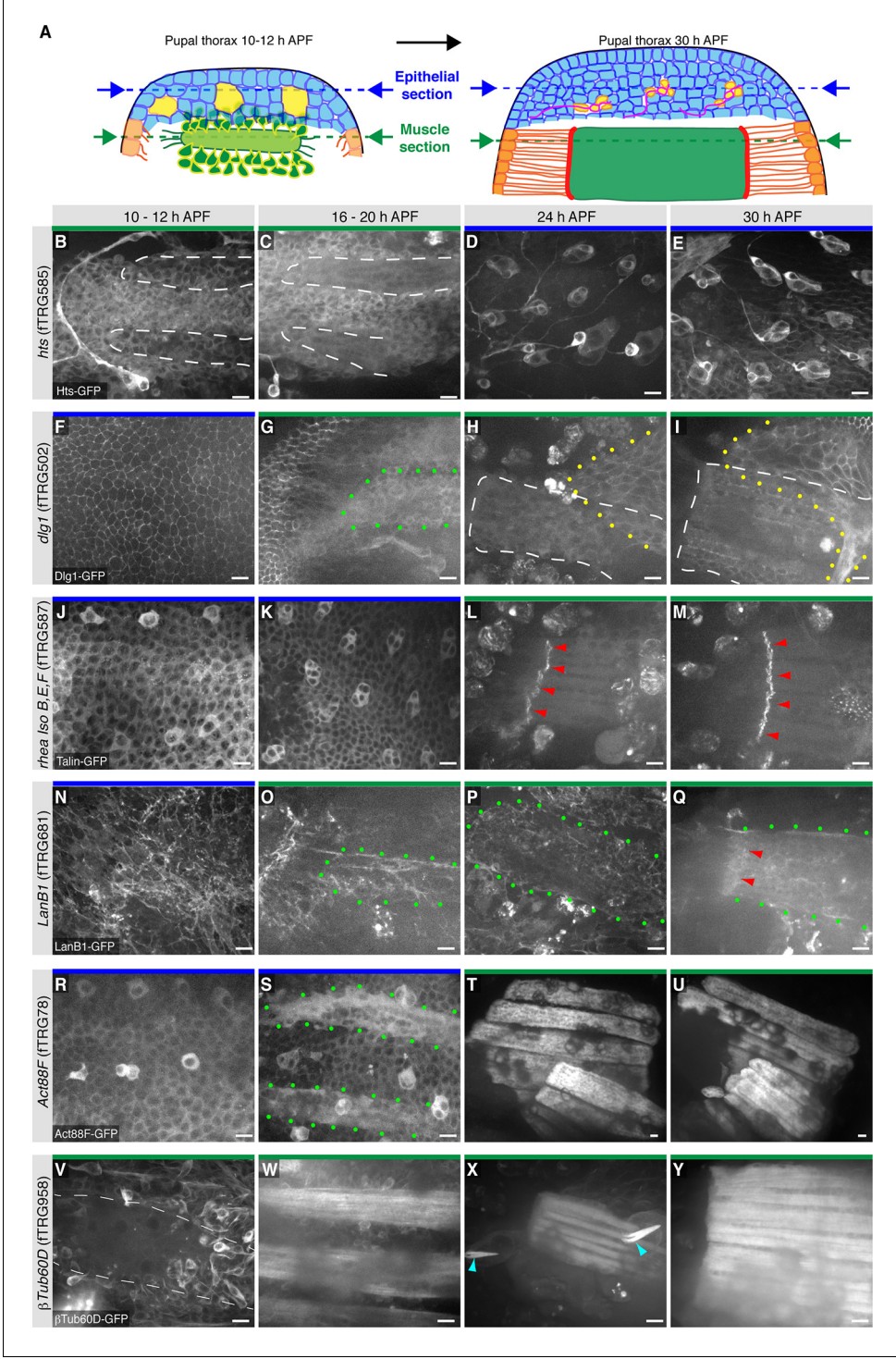

**Figure 8.** Live imaging of fTRG tagged proteins in living pupal thorax. (**A**) Schematic drawing of a 10–12 hr (left) and a 30h pupal thorax (right). The developing epidermis is shown in blue, with the SOP precursors in yellow (developing neurons in red), the differentiating tendons are shown in orange, the myoblasts and muscle fibers in green, and the muscle-tendon junction in red. The schematic positions of the optical sections through epithelium and muscles are indicated with blue and green dotted lines, respectively. (**B-Y**) Live imaging of pupal thoraces at the indicated stages acquired with a spinning disc confocal (except S and T, which were acquired with a two-photon microscope). Blue bars above the image indicate epithelial sections and green bars indicate muscle sections (as explained in **A**). Hts-GFP is expressed in fusing myoblasts (**B**, **C**) and strongly in developing SOPs (**D**, **E**). Dlg1-GFP labels the epithelial junctions (**F**), internal muscle structures (green dots, **G**) and an unidentified

*Figure 8 continued on next page*

*Figure 8 continued*

additional developing epithelium (yellow dots, **H, I**). Talin-GFP is higher expressed in developing SOPs (**J, K**) and strongly localised to the muscle-tendon junction from 24 hr APF (red arrowheads, **L, M**). LanB1-GFP localises to the basal side of the developing epithelium (**N**) and surrounds the forming muscle fibers (green dots, **O-Q**) with a slight concentration at the muscle-tendon junction at 30 hr APF (red arrowheads, **Q**). Act88F-GFP weakly labels the developing epithelium, with a slight concentration in the SOPs until 20 hr APF (**R, S**) and very strongly marks the IFMs from 24 hr onwards (**T, U**). βTub60D-GFPis expressed in the fusing myoblasts (**V, W**) and also labels the microtubule bundles in the developing muscle fibers (**X, Y**) and hair cells of the developing sensory organs (light blue arrow heads in **X**). Scale bars indicate 10 μm.
The following figure supplement is available for figure 8:

**Figure supplement 1.** Live imaging during pupal development.

---

spinning disc-confocal. This enabled us to visualise single myoblast fusion events in developing IFMs of an intact pupa (*Video 5*, *Figure 8—figure supplement 1G-K*). The six dorsal longitudinally oriented IFMs develop from three larval template muscles to which myoblasts fuse to induce their splitting into six myotubes (*Fernandes et al., 1991*). Using high resolution imaging of the βTub60D-GFP line, we find that most myoblasts fuse in the middle of the developing myotube during myotube splitting, with prominent microtubules bundles located at the peripheral cortex of the splitting myotube (*Video 7*, *Figure 8—figure supplement 1R*). These prominent microtubules bundles are then relocated throughout the entire developing myotube (*Video 7*, *Figure 8—figure supplement 1S-V*). Taken together, these live imaging data suggest that many fTRG lines will be well suited for high resolution live imaging of dynamic subcellular protein localisation patterns in developing *Drosophila* organs. This will strongly expand the set of live markers available for research in flies.

## Fly transgeneOme library as bait for proteomics

For the proper composition, localisation and in vivo function of most protein complexes endogenous expression levels of the individual components are critical (*Rørth et al., 1998*; *Tseng and Hariharan, 2002*). Hence, the TransgeneOme library would be an ideal experimental set-up to purify protein complexes from different developmental stages using endogenous expression levels of the bait protein. In principle, all the small affinity tags (TY1, V5, FLAG) (*Figure 1—figure supplement 1*) can be used for complex purifications. The presence of precision and TEV cleavage sites even allow two-step purifications. For proof of principle experiments, we selected four tagged proteins as baits: Ilk, Dlg1, Talin and LanB1, and analysed two

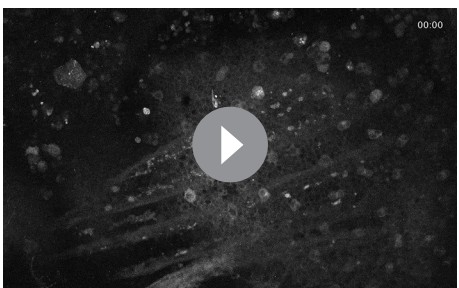

**Video 4.** Z-projection of a two-photon Video of an about 14 hr APF pupa expressing Act88F-GFP. A stack was acquired every 20 min for 19 hr. Expression of Act88F-GFP increases in the indirect flight muscles dramatically, thus contrast was reduced several times in course of the Video to avoid over-exposure. Video plays with 5 frames per second. Time is given in hh:mm.

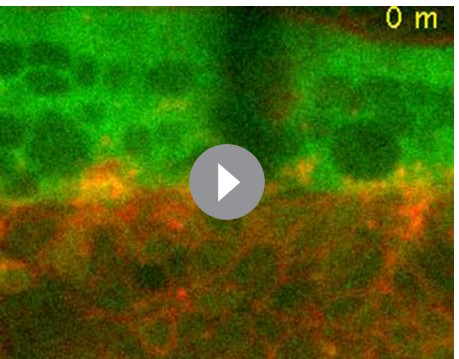

**Video 5.** Single plane of a spinning disc confocal Video of an about 14 hr APF old pupa expressing Act88F-GFP (green) in the flight muscle myotubes and *him*-GAL4; UAS-palm-Cherry in the myoblasts. An image stack was acquired every two minutes. Note the newly fused myoblasts acquired the GFP label within a single time interval (highlighted by green arrows). Video plays with 5 frames per second. Time is given in minutes.

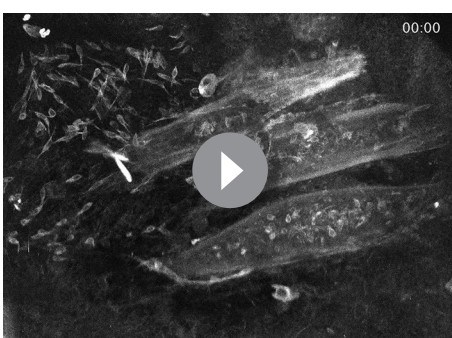

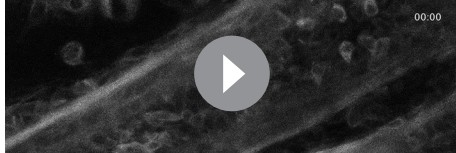

**Video 6.** Z-projection of a two-photon Video of an about 14 hr APF pupa expressing βTub60D-GFP. A stack was acquired every 20 min for 25 hr. Note the high expression of βTub60D-GFP in fusing myoblasts and the thick microtubules bundles in the developing flight muscles. Hair cells of the developing sensory organs also show strong expression, however, move out of the Z-stack over time. Video plays with 5 frames per second. Time is given in hh:mm.

**Video 7.** Single plane of a two-photon Video of an about 16 hr APF old pupa expressing βTub60D-GFP in myoblasts and the forming flight muscle myotubes. An image stack was acquired every two minutes for more than 3 hr. Note that single myoblasts can be followed during fusion. Most myoblasts fuse at the center of the myotube, which gradually splits into two myotubes. Video plays with 5 frames per second. Time is given in hh:mm.

different developmental stages. In each case we homogenised hundred 24 to 48 hr pupae and hundred adult flies per experiment and mixed the cleared lysate with a GFP antibody matrix to perform single step affinity enrichment and mass-spec analysis modified from the QUBIC protocol (*Hein et al., 2015*; *Hubner et al., 2010*; *Keilhauer et al., 2014*). Each affinity-enrichment was performed in triplicate and intensity profiles of all identified proteins were quantified in a label-free format by running all 30 purifications consecutively on the same Orbitrap mass-spectrometer and analysing the data with the MaxQuant software suite (*Cox and Mann, 2008*; *Cox et al., 2014*) (*Supplementary file 5*). Interestingly, enriching Ilk-GFP from both, developing pupae and adult flies, recovered the entire Ilk, PINCH, Parvin, RSU-1 complex (*Figure 9*), which had previously been purified in vitro from *Drosophila* S2 cells (*Kadrmas et al., 2004*) and mammalian cells (*Dougherty et al., 2005*; *Tu et al., 2001*) giving us confidence in our methodology. We also successfully enriched Talin-GFP from pupae or adults, however did not identify an obvious strong and specific binding partner (*Figure 9*, *Supplementary file 5*). In contrast, we identified Mesh as a novel interactor of Dlg1 from pupae and adult flies. Mesh colocalises with Dlg1 at septate junctions of the embryonic *Drosophila* midgut, however, a molecular interaction of both proteins was not established (*Izumi et al., 2012*). Finally, we purified the laminin complex by pulling on LanB1, which recovered LanB2 and LanA roughly stoichiometrically, both from pupae and adult flies, as had been found in cell culture experiments (*Fessler et al., 1987*), showing that extracellular matrix complexes can also be purified from in vivo samples with our methodology. In summary, these data demonstrate that interaction proteomics with the fly TransgeneOme library can confirm known interaction partners and discover novel in vivo complex members, making the system attractive for a variety of biochemical applications.

## Discussion

The TransgeneOme resource presented here adds a new powerful component to the arsenal of tools available to the *Drosophila* research community. It complements the genetic resources for gene disruption and localisation (*Buszczak et al., 2007*; *Lowe et al., 2014*; *Morin et al., 2001*; *Nagarkar-Jaiswal et al., 2015*; *Quiñones-Coello et al., 2007*; *St Johnston, 2012*; *Venken and Bellen, 2012*; *Venken et al., 2011*) with a comprehensive genome-scale library that does not suffer the biases of random mutagenesis. Analogously to the powerful MiMIC system (*Nagarkar-Jaiswal et al., 2015*; *Venken et al., 2011*), the TransgeneOme resource is versatile and can be adapted to the developments in tag chemistry and to various specialised applications. Although the resource is designed to study behaviour of proteins, it can for example be converted into a toolkit for live imaging of mRNAs. By designing a tagging cassette with an array of MS2 binding sites (*Forrest and Gavis, 2003*) the existing 'pre-tagged' TransgeneOme can be converted into an MS2-tagged TransgeneOme by a single liquid culture recombineering step in bacteria.

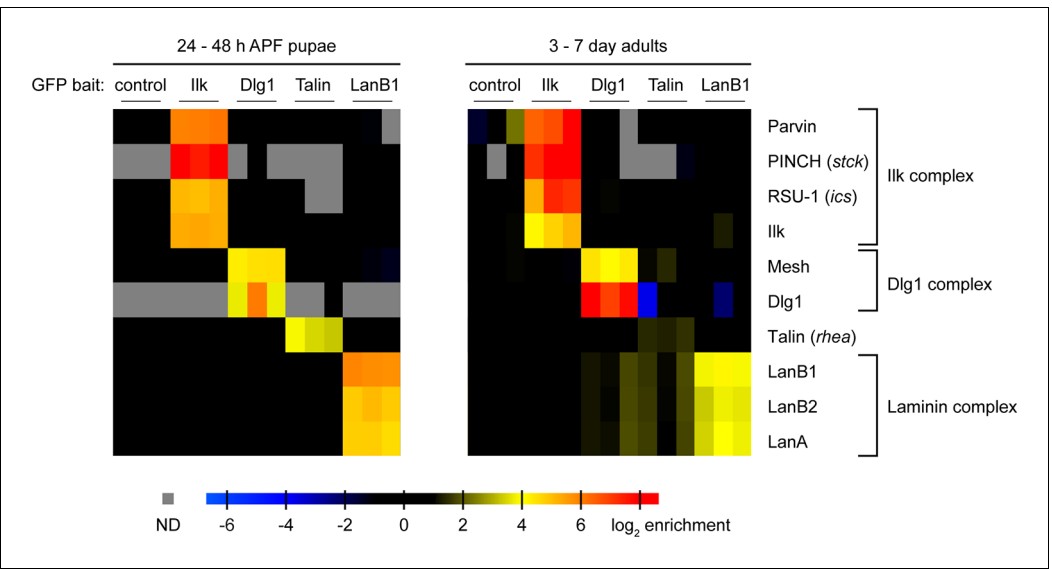

**Figure 9.** Proteomics with fTRG bait proteins. GFP-tagged Ilk, Dlg1, Talin, and LanB1, respectively, were affinity-enriched from protein extracts generated from whole pupae (left) or adult flies (right) using anti-GFP immunoprecipitation. A wild-type fly strain not expressing any GFP-tagged protein served as control. Proteins were quantified using mass spectrometry and the MaxLFQ label-free quantification algorithm in MaxQuant. Selected proteins are visualized by their enrichment factors in individual samples over the control (or simulated noise level, if not detected in the control). Specific interaction partners are characterised by the similarity of the quantitative profiles and co-enrichment with the respective bait proteins

However, any new TransgeneOme has to be transformed into flies, and this process still represents a significant bottleneck. We present here an optimised protocol for transgenesis of fosmid size clones into *Drosophila melanogaster* that was adapted from a previous large-scale transgenesis project (*Venken et al., 2010*). It took three years and four dedicated technicians to generate the 880 fly lines presented in this study. Although, the systematic transgenesis is a continuing process in our laboratories, the value of the TransgeneOme collection is highlighted by the fact that any specific set of genomically tagged gene clones is now available. These can be efficiently transformed by in house transgenesis of *Drosophila* labs around the world using the optimised protocol presented here.

One caveat of designed expression reporters is the necessity to place the tag into a defined position within the gene model. We chose to generate our 'pre-tagged' collection at the most commonly used C-terminus predicted by the gene model, thus labelling most isoforms. In some cases, a tag at the C-terminus will inactivate the protein, however such a reagent can still be useful for visualising the protein, although the result needs to be treated with care. This has been demonstrated for a number of sarcomeric protein GFP-traps, some of which lead to lethality when homozygous, yet result in interesting localisation patterns when heterozygous (*Buszczak et al., 2007*; *Morin et al., 2001*). Similarly, we found an interesting localisation pattern for BM40-SPARC-GFP despite being non functional. In some cases the tag may result in a mis-localisation of the tagged protein, compared with the untagged endogenous one, in particular when the tag interferes with protein function. For particular genes, it will be useful to tag differential protein isoforms, which in some cases can be done by tagging alternative C-termini, as shown here for *Mhc* and *rhea*. However, tagging a particular isoform requires a very informed construct design, which cannot easily be automated at the genome scale.

A functional GFP-tagged gene copy, as present in our fTRG lines, can also serve as a 'conditional' allele, when crossed into the mutant background and combined with deGradeFP (degrade Green Fluorescent Protein), an elegant method expressing a nanobody against GFP, coupled with a degradation signal in a tissue- and stage-specific manner (*Caussinus et al., 2011*; *Neumuller et al., 2012*). As the expression of the nanobody can be turned on or off, it is also possible to reversibly

remove the tagged protein, as recently shown for the GFP-tagged MiMIC lines (*Nagarkar-Jaiswal et al., 2015*). This introduces yet an other level of experimental manipulation, directly controlling protein levels at a given developmental stage.

Genome engineering is experiencing a tremendous growth with the introduction of CRISPR/Cas technology, and it will be only a matter of time before a larger collection of precisely engineered fusion proteins at endogenous loci will become available in flies. However to date, such examples are still limited to a few genes (*Baena-Lopez et al., 2013*; *Gratz et al., 2014*; *Port et al., 2014*; *Zhang et al., 2014*), which had been carefully picked and were individually manipulated with custom-designed, gene-specific tools. It remains to be tested which proportion of such engineered loci will be fully functional and thus potentially superior to the fTRG collection. However, having a transgenic third allele copy, as is the case in our TransgeneOme collection, might even be advantageous, if the tagging interferes with protein function, because the TransgeneOme lines still retain two wild-type endogenous gene copies. In some cases, addition of GFP might destabilise the protein, regardless of N- or C-terminal fusion, as recently shown for the Engrailed protein (*Sokolovski et al., 2015*). However, our ability to detect the protein product in the vast majority of our tagged lines argues that this could be a relatively rare, gene-specific phenomenon. Nevertheless, caution should be taken with respect to protein turnover dynamics of any tagged protein.

An additional advantage of our transgenic resource, independent of whether or not the gene is tagged, is that it can be used to rescue a classic genetic mutation and thus formally demonstrate that any observed phenotype is caused by the mutation in the studied gene. This cannot easily be done when modifying the endogenous gene copy by a MiMIC insertion or a CRISPR induced mutation. Thus, our resource complements previously published collections of genomic constructs (*Ejsmont et al., 2009*; *Venken et al., 2009*).

Together, the FlyFos library, the fly TransgeneOme library and the fTRG collection of strains, enable genome-scale examination of expression and localisation of proteins comparable with the high-throughput mRNA in situ screens (*Tomancak et al., 2002*; *2007*). Our data for tagged Oscar protein show that fosmid-based reporters can in principle recapitulate all aspects of gene expression regulation at transcriptional and post-transcriptional levels. It will be particularly interesting to combine the spatial expression data of mRNAs with that of proteins. Since many transcripts show subcellular localisation patterns in various developmental contexts (*Jambor et al., 2015*; *Lécuyer et al., 2007*), the question arises whether RNA localisation generally precedes localised protein activity. Systematic examination of protein patterns expressed from localised transcripts in systems such as the ovary will provide a genome-scale overview of the extent and functional role of translational control. At the tissue level, the patterns of mRNA expression may be different from the patterns of protein expression, for example due to translational repression in some cells or tissue specific regulation of protein stability, as shown here for the Corolla protein. The combined mRNA and protein expression patterns may therefore uncover a hidden complexity in overall gene activity regulation and the fTRG lines will help to reveal these combinatorial patterns in a systematic manner.

The fTRG lines faithfully recapitulate gene expression patterns in ovaries, embryos, larvae, pupae and adults suggesting that they can be used to visualise proteins in every tissue during the life cycle of the fly. This includes adult tissues such as the flight or leg muscles, which thus far had not been subjected to systematic protein expression and localisation studies. However, due to their size and the conservation of the contractile apparatus, these tissues are particularly attractive to study with this new resource. In general, antibody or FISH (Fluorescent In Situ Hybridisation) stainings with a single standard anti-tag reagent are easier to optimise, compared to antibody stainings or mRNA in situ hybridisations with gene-specific antibodies/probes. This simplicity makes it possible to explore the expression of the available genes across multiple tissues, as has been done for the *rab* collection (*Dunst et al., 2015*). Such an approach is orthogonal to the collections of expression data generated thus far, in which many genes were examined systematically but only at particular stages or in certain tissues, i.e. embryos or ovaries (*Jambor et al., 2015*; *Lécuyer et al., 2007*; *Tomancak et al., 2007*). We are confident that the analysis of regularly studied as well as less explored *Drosophila* tissues will be stimulated by the fTRG collection.

When protein expression levels are sufficiently high, the fusion proteins can be visualised by live imaging approaches in intact animals. It is difficult to estimate the absolute expression levels required for live visualisation, as this depends on the imaging conditions, the accessibility and transparency of the tissue and importantly on the observed protein pattern. A strongly localised protein

can result in a very bright local signal, such as Talin or Ilk at the muscle attachment sites or Gsb in the neuroblast nuclei, compared to a protein homogenously distributed throughout the entire cell. Given optimal imaging conditions, we estimate conservatively that about 50% of the tagged proteins can be visualised live, if they are expressed in tissues accessible to live imaging. In particular for tissues, such as the adult legs, antennae or the adult fat body, which are difficult to dissect and stain without losing tissue integrity, these live markers should be enormously beneficial.

One important limitation for examining the pattern of protein expression is the accessibility of the tissue of interest for imaging. We have shown that light sheet microscopy can be used to image the dynamics of tagged protein expression throughout embryogenesis. We further demonstrated that two-photon microscopy can be applied to study protein dynamics during muscle morphogenesis in developing pupae. Other confocal or light sheet-based imaging paradigms could be adapted for in totoimaging of living or fixed and cleared specimen from other life cycle stages. Establishing standardized protocols for preparation, staining and imaging of *Drosophila* stages, isolated tissues and organs will be necessary to realise the full potential of the fTRG collection.

Protein interaction data in fly are available from a number of studies (*Formstecher et al., 2005*; *Giot et al., 2003*; *Guruharsha et al., 2011*). These results were generated using yeast two-hybrid, or over-expression in a tissue culture system, followed by affinity purification and mass spectrometric analysis. Despite high-throughput, these approaches face the problem that the interacting proteins might not be present at the same place within a cell, or not even co-expressed in a developing organism. This is circumvented by affinity purifications of endogenously expressed proteins, which thus far at genome-scale was only reported from yeast (*Gavin et al., 2002*; *Ho et al., 2002*; *Krogan et al., 2006*). In higher organisms, BAC-based systems, which are closely related to our fosmid approach, elegantly solved these issues, as shown by a recent human interactome study (*Hein et al., 2015*).

The collection of transgenic fTRG lines covers currently only about 10% of the available tagged TransgeneOme clones. Expanding the fly collection to include most genes of the genome and importantly characterising the expression of the tagged proteins by imaging in various biological contexts is best achieved by spreading the clones and transgenic lines amongst the community of researchers using *Drosophila* as a model system. Therefore, all transgenic lines are available from the VDRC stock collection and all TransgeneOme clones from Source Biosciences. Despite the expanding CRISPR-based genome engineering technologies, the fTRG collection will continue to be an important resource for the fly community, in particular, if the full functionality of certain fTRG lines has been demonstrated, as we did here for a selection of important developmental regulators. As with many genome-scale resources it is typically easier to produce them than to fully characterise and exploit their potential. Comprehensive generation of thousands of transgenes and their thorough analysis takes time; it took us 4 years to assemble the collection presented here. Development of protocols and techniques to image these collections of tagged lines and assembling open access databases to share the data needs to continue and will eventually become useful also for the characterisation of resources whose production began only recently.

Wangler, Yamamoto and Bellen convincingly argued that the *Drosophila* system remains an indispensable model for translational research because many essential fly genes are homologs of Mendelian disease genes in humans (*Wangler et al., 2015*). Yet, even after decades of research on fruit flies only about 2,000 of the estimated 5,000 lethal mutations have been investigated. Resources like ours will therefore provide essential functional information about gene expression and localisation in *Drosophila* tissues that can serve as a starting point for the mechanistic understanding of human pathologies and their eventual cures.

## Materials and methods

### TransgeneOme clone engineering

Fosmids were engineered as described previously (*Ejsmont et al., 2009*; *2011*), except for the inclusion of the 'pre-tagging' step in the genome-wide TransgeneOme set. All tagging cassettes were generated from synthetic DNA and cloned into R6K carrying plasmids, which require the presence of the *pir* gene product for replication (*Metcalf et al., 1996*). The *pir* gene is not present in the

FlyFos library host strain, thereby ensuring near-complete lack of background resistance in the absence of the correct homologous recombination event.

Details of the recombineering steps are as follows (*Figure 1B*): Step 1. The *E. coli* cells containing a FlyFos clone covering the gene locus of interest are transformed with the pRedFlp plasmid, containing the genes necessary for the homologous recombination and the Flp recombinase under independently inducible promoters. Step 2. Next, a 'pre-tagging' cassette carrying an antibiotic resistance gene (NatR, nourseothricin resistance) surrounded by regions of homology to all specific tagging cassettes (*Figure 1—figure supplement 1*) and flanked by gene-specific homology arms is electroporated as linear DNA fragment produced by PCR. By combination of induced (L-rhamnose) pRedFlp homologous recombination enzyme action and strong selection with a cocktail of three antibiotics (one to maintain the fosmid (chloramphenicol, Cm), one to maintain the pRedFlp (hygromycin, Hgr) and nourseothricin (Ntc) to select for the inserted fragment) the electroporated linear 'pre-tagging' fragment becomes inserted in front of the STOP codon of the gene of interest. Step 3. The 'pre-tagging' cassette is exchanged for a cassette of the chosen tag coding sequence including an FRT-flanked selection / counter selection marker (rpsL-neo). This cassette is now universally targeting the homologous sequences shared by the tagging and pre-tagging cassettes and is produced in bulk by restriction enzyme-mediated excision from a plasmid. Note that in this way, no PCR-induced mutations can be introduced at this step. Step 4. Upon Flp induction (with anhydrotetracycline), the rpsL-neo cassette is excised, leaving a single FRT site, positioned in frame after the tag coding sequence. In this way, the endogenous STOP codon and the 3'-UTR of the tagged gene are used. Step 5. Finally, the recombineering plasmid is removed from the cells containing the engineered fosmids by inhibition of its temperature sensitive origin of replication and release from Hgr selection. The cells are plated on a selective chloramphenicol agar plate, from which a single colony is picked and further validated.

## NGS-based validation of the TransgeneOme clones

For NGS-based validation of the TransgeneOme library single colonies for each TransgeneOme clone were picked into 96-well plates, grown to saturation and the individual wells of all 96-well plates were pooled into 8 row and 12 column pools. Fosmid DNA was isolated from these pools and barcoded mate pair fragment libraries were prepared using the Nextera matePair library preparation chemistry from Illumina. The library was size selected through agarose gel isolation of approximately 3 kb fragments and sequenced on HiSeq 2500 (Illumina), with paired-end read lengths of 100 bp. Adapters and low quality sequences were trimmed with Trimmomatic0.32. (parameters: ILLUMINA-CLIP:NexteraPE-PE.fa:2:30:10 LEADING:3 TRAILING:3 SLIDINGWINDOW:4:15 MINLEN:36). To detect un-flipped fosmid sequences (where the FLP-mediated excision of selection cassette failed), the read pairs were mapped with Bowtie2 (*Langmead and Salzberg, 2012*) against the un-flipped tag sequence and the genome. If any read of the mate of the pair mapped to the un-flipped sequence, while the second mate mapped to the genome consistent with the estimated mate pair insert size of 3000 bp ± 1000 bp, the fosmid was flagged as un-flipped and was not further analysed. To identify mutations in the tag and in the immediate genomic surrounding (± 1000 bp), the NGS reads were mapped against the fosmid references that included the flipped tag. The Bowtie2 was set to report only hits where both reads of the pairs map concordantly to the insert size in the tag and in the genome (parameters: -I 2200 -X 3700 –rf –no-discordant –no-unal –no-mixed). PCR duplicated read pairs were removed using samtools1.1 *rmdup* (*Li et al., 2009*). Mutations were identified by utilising SNP calling implemented in FreeBayes (*Garrison and Marth, 2012*) using the standard filters and *vcffilter* to eliminate reported SNPs with scores < 20. Finally, in the last step, the information of the row and column pools were compared and summarized using a custom C-program that read the results of the SNP calling and the Bowtie mappings and counted the coverage for each read pair anchored in tag sequence with at least 20 bp. To correct for random PCR or sequencing errors the reported SNPs were compared for the row and column pools of each fosmid and SNPs occurring in both pools with coverage of 3 or more reads were considered as real.

## *Drosophila* stocks and genetic rescue experiments

Fly stocks were maintained using standard culture conditions. All crosses were grown at 25°C unless otherwise noted. Most of the fly mutant or deficiency strains for the rescue experiments were

obtained from the Bloomington *Drosophila* Stock Center and if located on X or 2$^{nd}$ chromosome crossed together with the respective fTRG line. If the mutant gene was located on the 3$^{rd}$ chromosome, it was recombined with the fTRG insertion. Rescue was generally tested in trans-heterozygotes as indicated in *Table 2*. The rescue for 6 genes (*bam, fat, mask, rap, RhoGEF2* and *yki*) was done by others, who communicated or published the results (*Table 2*). For the rescue of flightlessness a standard flight test was used (*Schnorrer et al., 2010*).

## Generation of transgenic fly TransgeneOme (fTRG) lines

Most TransgeneOme fosmid clones were injected into the y[1], w[*], P{nos-phiC31int.NLS}X; PBac{y +-attP-3B}VK00033 (BL-32542). This stock has white eyes and no fluorescent eye markers, which would interfere with screening for the red fluorescent eye marker used in the FlyFos clones (*Ejsmont et al., 2009*). A few fosmid clones were also injected into y[1], w[*], P{nos-phiC31int.NLS} X; PBac{y+-attP-3B}VK00002, with the attP site located on the 2$^{nd}$ chromosome. The *osk-GFP* fosmid was injected into attP40. Please note that all fTRG lines contain the strong 3xP3-dsRed marker (*Ejsmont et al., 2009*). This is an eyeless derived promoter fragment resulting in dsRed expression in the developing eye and in the brain. This needs to be taken into account when working with the developing or adult brain.

### Detailed injection protocol (adapted from *Venken et al., 2010*)

a. A) Bacterial culture of fosmid clones: 1. Inoculate 2 ml LB-medium plus chloramphenicol (Cm 25 µg / ml) with fosmid clone and grow overnight at 37°C. 2. Dilute to 10 ml (9 ml LB-medium + Cm and 1 ml bacterial culture) and add 10 µl 10% arabinose (final concentration 0.01% ) to induce the fosmid to high copy number. 3. Grow at 37°C for 5 hr and collect the pellet by 10 min centrifugation at 6000 rpm. Pellet can be stored at -20°C.

b. Preparation of fosmid DNA: Use the HiPure Plasmid Miniprep Kit from Invitrogen (order number: K2100-03) according to the supplied protocol (MAN0003643) with following modifications: before starting: pre-warm the elution buffer (E4) to 50°C; step 4: incubate the lysate for 4 min at room temperature; step 5: incubate 4 min on ice before centrifuging at 4°C for 10 min; step 8: add 850 µl elution buffer (pre-warmed to 50°C) to the column; step 9: add 595 µl isopropanol to the elution tube, centrifuge 20 min at 4°C; wash pellet with 800 µl 70% ethanol; centrifuge for 2 min; step 12: air dry the pellet for 4 min. Add 20 µl EB-buffer (Qiagen) to the pellet and leave at 4°C overnight to dissolve without pipetting to avoid shearing of the DNA. Do not freeze the DNA. Adjust the concentration to 250 ng / µl and centrifuge 5 min at full speed before injections. Do not inject DNA older than one week.

c. Embryo injections: Collect young embryos (0–30 min) on an agar plate, bleach away the chorion, wash and collect the embryos on a cellulose filter (Whatman 10409814). Align the embryos, transfer them to a glued slide and dry them with silica gel for 10–15 min (Roth T199.2). Cover the embryos with Voltalef 10S oil (Lehmann & Voss) and inject the prepared fosmid DNA using a FemtoJet set-up (Eppendorf 5247). The injected DNA should be visible within the embryo. Incubate the injected embryos for 48 hr at 18°C in a wet chamber and collect the hatched larvae with a brush. Cross the surviving mosaic adults individually to *y, w* males or virgins.

## Immuno-stainings and Western blotting

Ovaries: sGFP-protein detection and antibody co-stainings of egg-chambers was done as previously described (*Dunst et al., 2015*). Detection of the *oskar-GFP* mRNA was performed with a *gfp*-antisense probe (*Jambor et al., 2014*) and co-staining of *osk* mRNA and Osk protein was done as previously described (*Jambor et al., 2011*) using a *gfp*-antisense probe and a rabbit anti-GFP antibody (1:1000, ThermoFisher). Rabbit anti-Osk was used 1:3000 (gift on Anne Ephrussi), mouse anti-Grk was used 1:100 (DSHB).

Adult thoraces: Antibody stainings of adult thoraces, including flight, leg and visceral muscles, were done essentially as described for adult IFMs (*Weitkunat and Schnorrer, 2014*). Briefly, thoraces from young adult males were fixed for 15 min in relaxing solution (20 mM phosphate buffer, pH 7.0; 5 mM MgCl$_2$; 5 mM EGTA, 5 mM ATP, 4% PFA) + 0.5% Triton X-100, cut sagittally with a sharp microtome blade and blocked for 1 hr at room temperature with 3% normal goat serum in PBS-0.5% Triton X-100. The samples were stained with primary antibodies overnight at 4°C, rabbit anti-GFP

1:2000 (Amsbio); mouse anti-Futsch 1:100, mouse anti-Dlg1 clone 4F3 1: 500, mouse anti-Prospero 1:30 (all Hybridoma Bank, DSHB), mouse anti-ATP5a 1:500 (Abcam clone 15H4C4), rabbit anti-Fln (gift of Jim Vigoreaux), rabbit anti-Kc cell Laminin H329 1:2000 (gift of Stefan Baumgartner), rabbit anti-Mlp84B 1:500 (gift of Kathleen Clark), mouse anti-Mhc 1:100 (gift of Judith Saide), Mouse anti-Obscurin 1:500 (gift of Belinda Bullard) rabbit anti-Par6 1:400 (gift of Jürgen Knoblich), washed and incubated with secondary antibodies coupled to Alexa dyes and rhodamine-phalloidin or phalloidin-Alexa-660 (all from Molecular Probes). After washing, the samples were mounted in Vectashield containing DAPI. Images were acquired with a Zeiss LSM 780 confocal microscope and processed with Fiji (*Schindelin et al., 2012*) and Photoshop (Adobe).

Protein detection by Western blotting used standard procedures. 15 adult males were homogenised in 200 µl SDS buffer (250 mM Tris pH 6.8, 30% glycerol, 1% SDS, 500 mM DTT) and 5 µl were loaded per lane of a 10% SDS-PAGE gel. The Immobilon membranes (Millipore) were blocked with 10% milk powder and incubated with primary antibodies overnight (mouse anti-V5 1:10,000 (Invitrogen), mouse anti-Dlg1 1:10,000, rabbit anti-Mlp84B 1:20,000, rabbit anti-Fln 1:10,000). Detection used POD-coupled secondary antibodies (Jackson labs) and chemiluminescence (Millipore) using a LAS4000 detector system (FujiFilm).

## Live imaging

SPIM imaging of embryos: De-chorionated embryos of the appropriate age were embedded in 1% low melting point agarose and mounted into a glass capillary. Fluorescent microspheres (FY050 Estapor microspheres, Merck Millipore; 1:4000) were included in the embedding medium for multi-view registration. The embryos were imaged using the Zeiss Lightsheet Z.1 with a Zeiss 20x/1.0 water-immersion Plan Apochromat objective lens with 0.8x zoom at 25°C using 488 nm laser set at 4 mW. Five views were imaged using dual-sided illumination with Zeiss 10x/0.2 illumination lenses. A mean fusion was applied to fuse both illumination sides after acquisition using the ZEN software (Zeiss). The views were acquired at 72° angles with a stack size of 130 µm and a step size of 1.5 µm. Exposure time were 30 ms per slice. Each slice consists of 1920 x 1200 pixels with a pixel size of 0.29 µm and a bit depth of 16 bits. The light sheet thickness was 4 µm at the center of the field of view. The embryos were imaged from the onset of GFP expression (determined empirically) until late embryogenesis with a time resolution of 15 min. Multi-view processing of the dataset was carried out using the Fiji plugin for multi-view reconstruction (*Preibisch et al., 2009*; *Schmied et al., 2014*), which was executed on a high performance computing cluster (*Schmied et al., 2015*). The multi-view reconstruction was followed by multi-view deconvolution (*Preibisch et al., 2014*), for which the images were down sampled by a factor of two. Videos were extracted via the Fiji plugin BigDataViewer (*Pietzsch et al., 2015*).

The Gsb-GFP fTRG line was crossed with the H2Av-mRFPruby line (*Fischer et al., 2004*; *Preibisch et al., 2014*), the embryos of this cross were imaged using a 40x/1.0 water immersion Plan Apochromat lens from Zeiss with 1x zoom at 25°C at 17.5 mW of the 488 nm laser and 4 mW of the 561 nm laser. A single angle with dual-sided illumination was imaged. The stack size was 82.15 µm with a step size of 0.53 µm. Exposure time was 30 ms per slice. Each slice consisted of 1920 x 1920 pixels with a pixel size of 120 nm and a bit depth of 16-bit. The light sheet thickness was 3.21 µm at the center of the field of view. The embryos were imaged from early blastoderm onwards until late embryogenesis focusing on the head with a time resolution of 7 min.

Imaging of pupae: Staging and live imaging of the pupae were performed at 27°C. Live imaging of pupae at the appropriate stage was done as described previously (*Weitkunat and Schnorrer, 2014*). Briefly, the staged pupa was cleaned with a brush and a small observation window was cut into the pupal case with sharp forceps. The pupa was mounted on a custom-made slide and the opening was covered with a small drop of 50% glycerol and a cover slip. Z-stacks of either single time points or long-term time-lapse Videos were acquired using either a spinning disc confocal microscope (Zeiss, Visitron) or a two-photon microscope (LaVision), both equipped with heated stages.

## Proteomics

Per sample about hundred pupae or adult flies were snap-frozen in liquid nitrogen and ground to a powder. The powder was re-suspended and further processed as described in the quantitative BAC-GFP interactomics protocol (*Hubner et al., 2010*). In brief, 800 µl of lysate per sample were cleared by centrifugation. The cleared lysate was mixed with magnetic beads pre-coupled to anti-GFP

antibodies and run over magnetic micro-columns (both Miltenyi Biotec). Columns were washed, and samples subjected to in-column tryptic digestion for 30 min. Eluates were collected and digestion continued overnight, followed by desalting and storage on StageTips. Eluted peptides were analysed with an Orbitrap mass spectrometer (Thermo Fisher). Raw data were analysed in MaxQuant version 1.4.3.22 (*Cox and Mann, 2008*) using the MaxLFQ algorithm for label-free quantification (*Cox et al., 2014*). Interacting proteins were identified by the similarity of their intensity profiles to the respective baits (*Keilhauer et al., 2014*). Heat maps were plotted in the Perseus module of the MaxQuant software suite.

## Acknowledgements

Stocks obtained from the Bloomington *Drosophila* Stock Center (NIH P40OD018537) were used in this study. We are grateful to Belinda Bullard, Anne Ephrussi, Judith Saide, Jürgen Knoblich, Stefan Baumgartner, Kathleen Clark, Jim Vigoreaux and the DSHB for generously sharing antibodies and fly lines. We thank Franziska Friedrich for drawing the ovariole scheme used for *Figure 3A*. We thank Andreas Dahl from the Deep Sequencing Group at CRTD/BIOTECH, Dresden for the NGS library preparation and sequencing. We also thank the light microscopy facility and the computer department for assistance with imaging and data processing. We are grateful to Sandra Lemke, Aynur Kaya-Copur and Xu Zhang for help with the genetic rescue experiments and to Cornelia Schönbauer, Beatrice Laudenbach and Caroline Sonsteby for assistance during some of the adult and pupal imaging experiments. We thank the entire Schnorrer lab for helpful comments on this manuscript. We are particularly grateful to Reinhard Fässler and Herbert Jäckle for continuous support of this work. This work was funded by the EMBO Young Investigator Program (FS), the European Research Council under the European Union's Seventh Framework Programme (FP/2007-2013)/ERC Grant 310939 (FS), the European Research Council under the European Union's Seventh Framework Programme (FP/2007-2013)/ERC Grant 260746 (PT), European Commission GENCODYS (PT and HJ), Human Frontier Science Program (HFSP) RGY0093/2012 (PT and CS) and the Max Planck Society (PT, FS, MS, MM and EK).

## Additional information

### Competing interests

KV: Senior editor, *eLife*. MR: Reviewing editor, *eLife*. MS, PT, FS: Co-inventor of the fosmid and fly resource collections under the rules of the Max Planck Society. The other authors declare that no competing interests exist.

### Funding

| Funder | Grant reference number | Author |
|---|---|---|
| Max-Planck-Gesellschaft | | Mihail Sarov<br>Christiane Barz<br>Helena Jambor<br>Marco Y Hein<br>Christopher Schmied<br>Dana Suchold<br>Bettina Stender<br>Irene RS Ferreira<br>Katja Finkl<br>Nicole Plewka<br>Elisabeth Vinis<br>Elisabeth Knust<br>Matthias Mann<br>Pavel Tomancak<br>Frank Schnorrer |
| European Research Council | ERC Grant 260746 + ERC Grant 310939 | Pavel Tomancak<br>Frank Schnorrer |
| European Molecular Biology Organization | | Frank Schnorrer |

| Human Frontier Science Program | | Christopher Schmied Pavel Tomancak |
| --- | --- | --- |
| European Commission | GENCODYS | Helena Jambor Pavel Tomancak |

The funders had no role in study design, data collection and interpretation, or the decision to submit the work for publication.

## Author contributions

MS, Performed the liquid culture recombineering experiments, Data and contributed to the TransgeneOme database, Initiated the collaborative project and obtained dedicated funding, Conception and design, Acquisition of data, Analysis and interpretation of data, Drafting or revising the article; CB, Performed most of the genetic rescue experiments, Collected and analysed all imaging data in pupae and adults, Acquisition of data, Analysis and interpretation of data; HJ, Performed and analysed all experiments in ovaries, Acquisition of data, Analysis and interpretation of data; MYH, Performed the proteomics analysis and analysed the data, Acquisition of data, Analysis and interpretation of data; CS, Collected and analysed SPIM in toto images of embryos, Acquisition of data, Analysis and interpretation of data; DS, SH, EV, Performed the liquid culture recombineering experiments, Acquisition of data; BS, KF, NP, Performed the transgenesis of about half the fTRG lines, Acquisition of data; SJ, PK, SS, Data and contributed to the TransgeneOme database, Analysis and interpretation of data; VVK, RTK, AK, Performed the transgenesis of the other half of the fTRG lines, Acquisition of data; IRSF, Performed initial proof of principle fosmid experiments in adult flies, Contributed unpublished essential data or reagents; RKE, Constructed the tagging cassettes, Acquisition of data; EK, initiated the collaborative project and obtained dedicated funding, Conception and design; VH, Contributed analysis of Gsb-GFP cell tracking, Acquisition of data, Analysis and interpretation of data; MM, Performed the proteomics analysis and analysed the data, Analysis and interpretation of data; MR, Performed the transgenesis of the other half of the fTRG lines, Acquisition of data, Analysis and interpretation of data; KV, Performed the transgenesis of the other half of the fTRG lines, Conception and design, Acquisition of data, Analysis and interpretation of data; PT, Wrote the scripts to analyse Drosophila gene models, Initiated the collaborative project and obtained dedicated funding, Conception and design, Acquisition of data, Analysis and interpretation of data, Drafting or revising the article; FS, Performed the transgenesis of about half the fTRG lines, Collected and analysed all imaging data in pupae and adults, Performed the Western blots of adult extracts, Initiated the collaborative project and obtained dedicated funding, Conception and design, Acquisition of data, Analysis and interpretation of data, Drafting or revising the article

## Author ORCIDs

Mihail Sarov, http://orcid.org/0000-0003-2895-4087
Marco Y Hein, http://orcid.org/0000-0002-9490-2261
Vinay Vikas KJ, http://orcid.org/0000-0003-1328-9395
Aishwarya Krishnamoorthy, http://orcid.org/0000-0001-8245-3828
K VijayRaghavan, http://orcid.org/0000-0002-4705-5629
Pavel Tomancak, http://orcid.org/0000-0002-2222-9370
Frank Schnorrer, http://orcid.org/0000-0002-9518-7263

# Additional files

### Supplementary files

• Supplementary file 1. TransgeneOme constructs. Construct and clone names, tag locations, as well the sequencing validation data are listed for all TransgeneOme constructs generated. Sheet 1 lists the sGFP TransgeneOme (TY1-sGFP-V5-BLRP-FLAG tag, NGS sequenced), sheet 2 the TY1-sGFP-FLAG pilot set clones (junctions Sanger sequenced, only exact matches are counted as verified), sheet 3 the TY1-T2A-sGFPnls-FLAG pilot set clones (entire tag Sanger sequenced) and sheet 4 the TY1-sGFP-V5-BLRP-FLAG pilot set clones (entire tag Sanger sequenced). Sheet 5 summarises all verified tagged genes in these sets.

• Supplementary file 2. fly TransgeneOme lines (fTRG) lines. Table listing all 880 transgenic FlyFos (fTRG) lines, with fTRG numbers, construct and clones names, as well as nature of the tag and the used landing site. The second sheet compares the genes tagged by the fTRG lines to the available GFP gene trap lines. 765 genes are only found in the TransgeneOme resource.

• Supplementary file 3. fTRG expression in ovaries. Table listing the expression patterns for 115 fTRG lines in ovaries. Expression was detected in 94 lines by anti-GFP antibody stainings. Cell type specific expression and subcellular localisations were monitored for these lines.

• Supplementary file 4. fTRG expression in the adult thorax. Table listing the expression pattern for 121 fTRG lines in adult thoraces. Expression was detected in 101 lines by anti-GFP antibody stainings. Cell type specific expression and subcellular localisations were monitored for these lines.

• Supplementary file 5. Proteomics quantification. Quantitative mass spectrometry values of all detected protein obtained with the MaxQuant software suite for all the GFP-enrichment experiments are listed.

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
