## [Decision Letter]

Thank you for submitting your work entitled "A genome-wide resource for the analysis of protein localisation in *Drosophila*" for consideration by *eLife*. Your article has been reviewed by three peer reviewers, one of whom, Hugo Bellen, is a member of our Board of Reviewing Editors. The evaluation has been overseen by Hugo Bellen as Reviewing Editor and Detlef Weigel as the Senior Editor.

The reviewers have discussed the reviews with one another and the Reviewing editor has drafted this decision to help you prepare a revised submission.

Summary:

This manuscript details the generation of about 10,000 carboxyterminal (C-terminally) tagged genes present in fosmid clones generated by Ejsmont et al. (2011). You use liquid recombineering to efficiently tag thousands of genes. Although some clones have mutations it seems that a collection of more than 9,000 tagged genes should be very useful. You then transform ~900 fosmids into flies, and show that about 2/3 of the genes are functional and are able to rescue mutations in the corresponding genes, based on 46 tested genes. You also document expression patterns in various tissues and at various stages using still and live imaging for 207 of the transgenes. Finally, you perform immunoprecipitations followed by Mass-spec to show that the transgenic animals can be used to identify the protein partners for some of the proteins tested.

The resource described provides a much needed tool for protein analysis but information from these numerous images is limited by both available rescue data for the imaged lines and lack of counterstaining using known markers for comparison. Hence, there are a few experiments that the three reviewers request that should be included.

Specifically, you should show with several antibody stains that the endogenous proteins and tagged proteins colocalise. Also, you should perform a few Western blots to show how the levels of tagged proteins compare to the endogenous proteins. The reviewers suggest that this be done for at least five different genes. Please do not biasthe selection. Please pick randomly antibodies and do colocalisation studies to assess if the protein distributions match or not. Minor issues include that you should carefully compare the localisation of the tagged proteins in the transgenic lines to those proteins for which good antibodies are available, and directly examine subcellular localisation by double-staining with antibodies against the protein and anti-GFP. You could also perform a direct comparison of the C-terminally tagged proteins that you have made to proteins tagged internally by MiMIC GFP (Nagarkar-Jaiswal et al.), and see if there is any overlap between your transgenic lines and the lines made by the MiMIC RMCE project.

In general, this is a very valuable and important resource and we support publication of this manuscript in *eLife* as a resource paper. Every *Drosophila* lab will likely use these reagents. The experiments to generate the tagged gene plasmids were very successful. The tag strategy seems reasonable, and many of the tagged proteins (although perhaps not as many as they would argue) are likely to be able to execute their normal functions. Thank you for generating this resource.

---

## [Author Response]

*[…] Specifically, you should show with several antibody stains that the endogenous proteins and tagged proteins colocalise. Also, you should perform a few Western blots to show how the levels of tagged proteins compare to the endogenous proteins. The reviewers suggest that this be done for at least five different genes. Please do not biasthe selection. Please pick randomly antibodies and do colocalisation studies to assess if the protein distributions match or not. Minor issues include that you should carefully compare the localisation of the tagged proteins in the transgenic lines to those proteins for which good antibodies are available, and directly examine subcellular localisation by double-staining with antibodies against the protein and anti-GFP. You could also perform a direct comparison of the C-terminally tagged proteins that you have made to proteins tagged internally by MiMIC GFP (Nagarkar-Jaiswal et al.), and see if there is any overlap between your transgenic lines and the lines made by the MiMIC RMCE project.*

We are very happy that the reviewers univocally agree on the broad value of our resource and on the quality of the presented data. As suggested by the reviewers, we have now included double stainings with anti-GFP antibodies and antibodies against the investigated protein for 9 different proteins, for which we had good antibodies available. We tried to select a representative set of proteins for which we had antibodies available and had used them already in the above expression analysis.

Additionally, we provide now antibody stainings for these 9 proteins in wild-type tissues, as not for all proteins the protein localisation was described in the literature. The tested proteins include Osk and Grk in ovaries, now presented in new Figure 3—figure supplement 1, and LanA, LanB1 (not shown), Par6, Mlp84, Mhc, Fln, Unc-89 (Obscurin) and Dlg1, now presented in new Figure 7. In all cases, we see a high degree of overlap between tagged protein and endogenous protein, demonstrating that the tagged proteins localise together with the endogenous proteins. Additionally, we confirmed a similar localisation pattern also in wild-type tissue ruling out the possibility that the tagged proteins induce an ectopic pattern. Only for the highly expressed long Mhc-GFP isoforms we do observe a slight difference of Mhc localisation in IFMs compared to the untagged Mhc in wild-type. This is most likely due to a slight myofibril phenotype caused by the Mhc-GFP, (these tagged flies cannot fly). We included this statement in the text.

These results are in good agreement with published large-scale tagging datasets that compared antibody stainings to GFP-tagged localisation patterns and find a co-localisation for 80% of the C-terminally tagged proteins. In the remaining cases it is unclear which method is more reliable (Stadler et al. Nat Meth 2013).

We now cite this paper in the Results and the Discussion, as the data presented in the paper also favour C-terminal over N‐terminal tagging.

Our double stainings suggest that also the levels of the tagged proteins are comparable to the endogenous proteins, which is in line with other systematic studies using large BAC clones for tagging,

e.g. demonstrated with Western blots in HELA cells (Hein et al. Cell 2015).

To directly address the reviewers’ concerns we have performed Western blots using total protein extract from adult males and probing with specific antibodies to observe the 40 kDa shift of the tagged protein compared to the endogenous protein. We were able to reliably do this for three different proteins, from which we could reliably detect the endogenous protein as clear band from total wild-type extract. Of course, one could always do more. However, we believe the examples make the case that the levels can be similar, as we show for 2 of 3 cases, or lower, as we show for the third example Mlp84B. These new data are now presented in Figure 7.

*In general, this is a very valuable and important resource and we support publication of this manuscript in eLife as a resource paper. Every Drosophila lab will likely use these reagents. The experiments to generate the tagged gene plasmids were very successful. The tag strategy seems reasonable, and many of the tagged proteins (although perhaps not as many as they would argue) are likely to be able to execute their normal functions. Thank you for generating this resource.*

Thank you for your support to publish this paper and for the very constructive review process! We highly appreciated it.

The clones have been meanwhile deposited at Source BioScience and can be ordered by the scientific community. The price per clone has been set by the company initially rather high. Based on the feedback from the community, we re-negotiated the price, which is now at the level of €120 (excluding shipping) per clone depending on the country of the customer. For comparison, untagged BAC clones are available from BACPAC Resources (a non-profit organisation as we understand) for $80 (excluding shipping). Thus the price of the tagged clones is double that of an un-tagged version. Although we would like to see the resource available free of charge, we think that this is a fair price from a commercial supplier who will offer additional added value services (i.e. whole collection, custom sets, etc.).